# Evaluation of Himawari-8 surface downwelling solar radiation by ground-based measurements

Alessandro Damiani[1], Hitoshi Irie[1], Takashi Horio[1], Tamio Takamura[1], Pradeep Khatri[2], Hideaki Takenaka[3], Takashi Nagao[3], Takashi Y. Nakajima[4], Raul R. Cordero[5]

5  [1]CEReS, Chiba University, Chiba, 263-8522, Japan
[2]Center for Atmospheric and Oceanic Studies, Tohoku University, Sendai, 980-8578, Japan
[3]Earth Observation Research Center, JAXA, Tsukuba, 305-0047, Japan
[4]Research and Information Center, Tokai University, Tokyo, 151-0063, Japan
[5]Department of Physics, Santiago University, Santiago de Chile, 8320000, Chile

*Correspondence to*: Alessandro Damiani (alecarlo.damiani@gmail.com)

**Abstract.** Observations from the new Japanese geostationary satellite Himawari-8 permit quasi-real-time estimation of global shortwave radiation at an unprecedented temporal resolution. However, accurate comparisons with ground truthing observations are essential to assess their uncertainty. In this study, we evaluated the Himawari-8 global radiation product 15 AMATERASS using observations recorded at four SKYNET stations in Japan and, for certain analyses, from the surface network of the Japanese Meteorological Agency in 2016. We found that the spatiotemporal variability of the satellite estimates was smaller than that of the ground observations; variability decreased with increases in the time step and spatial domain. Cloud variability was the main source of uncertainty in the satellite radiation estimates, followed by direct effects caused by aerosols and bright albedo. Under all-sky conditions, good agreement was found between satellite and ground-20 based data, with a mean bias in the range of 20–30 W/m$^2$ (i.e., AMATERASS overestimated ground observations) and a root mean square error (RMSE) of approximately 70-80 W/m$^2$. However, results depended on the time step used in the validation exercise, on the spatial domain and on the different climatological regions. In particular, the validation performed at 2.5 min showed largest deviations and RMSE values ranging from about 110 W/m$^2$ for the mainland to a maximum of 150 W/m$^2$ in the subtropical region. We also detected a limited overestimation in the number of clear-sky episodes, particularly at the 25 pixel level. Overall, satellite-based estimates were higher under overcast conditions, whereas frequent episodes of cloud-induced enhanced surface radiation (i.e., measured radiation was greater than expected clear-sky radiation) tended to reduce this difference. Finally, the total mean bias was approximately 10–15 W/m$^2$ under clear-sky conditions, mainly because of overall instantaneous direct aerosol forcing efficiency in the range of 120–150 W/m$^2$ per unit of aerosol optical depth (AOD). A seasonal anti-correlation between AOD and global radiation differences was evident at all stations and was also 30 observed within the diurnal cycle.

# 1 Introduction

Following the Tōhoku Earthquake of March 2011, governmental policy in Japan stimulated a broader use of renewable energy sources. However, full integration of the natural energy resource is still prevented by its variability. In particular, the instability of solar power caused by cloudiness is a critical issue for suppliers. One way to overcome this variability is to develop an appropriate control system for electric power generated by solar radiation and dispensed into the power grid system, i.e., an energy management system (EMS). However, to implement such an EMS system, surface solar irradiance data must be supplied as accurately as possible.

Although ground instruments provide the most reliable monitoring results, they demand special maintenance and are usually sparse. By contrast, satellites can be easily used to characterise large regions, albeit with limited temporal and spatial resolution. To date satellite-based estimates of surface global solar radiation are routinely performed by different algorithms which exploit the images recorded by geostationary satellites, such as the Meteosat Second Generation (MSG) and the Geostationary Operational Environmental Satellites (GOES), to relate satellite reflectance to cloud opacity. Among the available algorithms, we remind the Heliosat-2 method (Raschke et al., 1987; Rigollier et al., 2004; Blanc et al., 2011), the State University of New York (SUNY) model (Perez et al., 2002), the Surface Insolation under Clear and Cloudy skies model (SICCS, Greuell et al., 2013) as well as more sophisticated radiative transfer models (e.g. Pinker and Laszlo, 1992) and algorithms based on neural network models (e.g., Takenaka et al., 2011; Taylor et al., 2016). Because of its unprecedented temporal resolution (up to 2.5 min), the new Japanese geostationary meteorological satellite Himawari-8 is expected shortly to attain a key position within this framework, drastically improving atmospheric research (Bessho et al., 2016) and forecasts of photovoltaic power generation (e.g. Ohtake et al., 2013, 2015).

Because punctual ground observations can only partially represent pixel area, depending on weather (Federico et al., 2017), topography (Gómez et al., 2016), and surface albedo (Damiani et al., 2012, 2013) conditions, accurate and extensive comparisons with ground truthing observations are essential to evaluating the accuracy of satellite-based estimates of solar radiation. This validation is performed by ground-based pyranometers usually on an hourly basis (e.g., Nottrott and Kleissi, 2010; Djebbar et al., 2012; Greuell et al., 2013; Federico et al., 2017). For example, Federico et al. (2017) recently evaluated estimates of surface solar irradiance over Italy based on the observations of MSG - Spinning Enhanced Visible and Infrared Imager (SEVIRI) and the SICCS model. They found that the root mean square error (RMSE) ranges between about 70 and 180 W/m$^2$ depending on the sky conditions. The study of Thomas et al. (2016a) highlighted that the HelioClim-3 database, derived from images acquired by the MSG, showed a large correlation coefficient (0.96) with the ground stations over Brazil. However, it tends to overestimate the observed surface solar radiation with most values comprised between 1 and 5 %. For the same database, Thomas et al. (2016b) showed a correlation ranging from ~0.94 to 0.98 and a relative RMSE between ~11 and 30 % with respect to some stations in Europe. The National Solar Radiation Database, based on visible

images from GOES and the SUNY model, showed an average overprediction of five percent when compared with surface stations in California due to its tendency to overestimate solar irradiance under cloudy conditions (Nottrott and Kleissi, 2010). A following evaluation of the same algorithm using GOES images over Canada showed an hourly mean bias error (MBE) of 5.6 W/m$^2$ and a RMSE of 86.5 W/m$^2$ when results were averaged over 18 stations (Djebbar et al., 2012).

In the recent literature, only a few validation exercises dealt with datasets at high temporal resolution (at 15 min, e.g. Ruf et al., 2016; Zo et al., 2016; Qu et al., 2017; Kosmopoulos et al., 2018). Zo et al. (2016) compared a satellite-based dataset with observational data from 22 solar sites of the Korean Meteorological Administration (KMA) over Korea and found an overall correlation coefficient of about 0.95. Satellite-based estimates at 15 min retrieved by the new Heliosat-4 method (Qu et al., 2017) were recently validated with observations from stations (mainly in Europe) of the Baseline Surface Radiation Network

(BSRN). In this study, an overestimation of the surface solar radiation in the range 2-32 W/m$^2$ (with the lower values over desert or semi-arid locations) and a RMSE between 74 and 94 W/m$^2$ were reported. A somewhat larger RMSE (115.9 W/m$^2$) for the same dataset has been shown for a single station in Germany (Ruf et al., 2016). For latitudes higher than 48 degrees, the correlation coefficient was found in the range of 0.90-0.94 while, for lower latitudes, in the range of 0.95-0.96 and this pointed to difficulty in the cloud discrimination at high satellite viewing angles. Moreover, Qu et al. (2017) showed a

positive correlation between cloud fraction and RMSE whose extent was likely affected by the specific climatological features at the different stations. Overall, in Europe the validation results were better for the summer months than for winter.

Although this validation is essential, there is also an increasing need to evaluate performance at even higher temporal and spatial resolutions (Perez et al., 2016), because rapid transients can be harmful to photovoltaic (PV) installations. Previous

studies have found that the correlation between sites decreases with distance and increases with temporal aggregation (Engeland et al., 2017). Although time series of spatially averaged irradiance fields generally resemble behaviour point measurements, their power spectra are strongly attenuated at higher frequencies and for large domains (Madhavan et al., 2017). Therefore, variation in spatial averages (or satellite pixels) and in point measurements are often poorly correlated at high frequencies (Perez et al., 2016). Indeed, a single pyranometer is usually representative of the overhead cloud structure

only for time resolutions larger than 1 h; at higher frequencies, sampling within the satellite pixel helps to reduce discrepancies with remotely sensed estimates (Nunez et al., 2013).

Overall, point measurements can deviate strongly from the spatial mean of a surrounding domain, and satellite/ground comparisons performed at medium/long time scales tend to smooth variability in the radiation, thus hiding important issues.

One of these is cloud-induced radiation enhancement (RE), which results in a surface radiation larger than the expected (simulated) clear-sky radiation (Gueymard et al., 2017). These events usually have only a small spatial footprint and, although quite frequent under broken cloud conditions, cannot be reproduced by satellite algorithms based on homogeneous parallel-plane and single-layer cloud models and may lead to problems in photovoltaic power stations. Indeed, earth-observing satellites cannot fully account for cloud inhomogeneity and this points to the necessity of a more extensive

implementation of 3D radiative transfer in the atmosphere with inhomogeneous clouds (Okamura et al., 2017; Iwabuchi, 2006). Even if such issues are masked by the broad time resolution used in validation exercises, as we will see, REs can introduce bias into the evaluation of satellite estimates.

Many additional factors can introduce a bias in the results of a validation exercise and make satellite-based estimates more uncertain (Polo et al., 2016). Among others, we remind the important role played by aerosols in reducing the surface solar irradiance, thus this is an important parameter to be accounted in the algorithms, especially in polluted regions or deserts under clear sky conditions (e.g. Qu et al., 2017). Then, it is worth to mention the negative effect of the complex morphology in mountainous regions which cannot be easily accounted due to the limited satellite spatial resolution and the local fast-changing weather conditions (Dürr et al., 2010; Urraca et al., 2017; Federico et al., 2017). Further, the negative impact of the high solar zenith angle (SZA) and satellite viewing zenith angle on the quality of the estimates must be also mentioned: the former can cause low clouds to be overshadowed by higher clouds while the latter leads to parallax effect in the clouds position (Polo et al., 2016; Qu et al., 2017). Finally, uncertainties in the surface albedo, especially over bright surfaces (e.g. snow or desert), can make cloud identification hard (e.g. Tanskanen et al., 2007).

In this study, we used ground-based observations to evaluate surface downwelling solar radiation estimates at high temporal resolution made by applying the EXtreme speed and Approximation Module multiple drive system (EXAM) algorithm (Takenaka et al., 2011) to Himawari-8 observations. Following an initial assessment of the spatiotemporal variability of the radiation fields, we focussed on effects caused by cloudiness, aerosols, and surface albedo. Although cloud-induced variability was expected to be the main source of uncertainty in satellite estimates of solar radiation, issues with albedo and aerosols caused further uncertainty under bright albedo and clear-sky conditions.

## 2 Data and Methods

### 2.1 Himawari-8 estimates of surface downwelling solar radiation

The Advanced Himawari Imagers (AHIs) aboard Himawari-8 acquire full-disk observations in 16 observation bands (three for visible, three for near infrared, and 10 for infrared wavelengths) every 10 min (and over Japan every 2.5 min), with a spatial resolution ranging from 0.5 to 2 km (Bessho et al., 2016). These observations allow quasi-real-time estimation of surface downwelling global shortwave (SW) radiation over Japan at a temporal resolution of 2.5 min and a nominal spatial resolution of 1 km using the EXAM algorithm (Takenaka et al., 2011). The algorithm is based on a fast-neural network, accurately reproducing the radiative transfer model. This satellite-based radiation data set, AMATERASS solar radiation, uses the Comprehensive Analysis Program for Cloud Optical Measurement (CAPCOM; Nakajima and Nakajima, 1995; Kawamoto et al. 2001) algorithm to retrieve cloud optical thickness and cloud-particle effective radius from Himawari-8 observations by a lookup table (LUT)–based approach under a homogeneous plane-parallel and single-layer cloud model. Additional input information included in EXAM, such as water vapour and ozone, was acquired from external data sets (e.g.,

the Japanese Reanalysis and OMI/Aura satellite), and surface albedo was computed from Himawari-8 observations using a statistical method. Although work is ongoing to include aerosol effect in the algorithm, the current version does not take aerosols into account. Many previous studies estimated the aerosol impact on surface solar radiation (e.g. Xia et al., 2007; Cachorro et al., 2008; Di Biagio et al., 2009, 2010; Papadimas et al., 2012; Huttunen et al., 2014). In particular, under clear

sky conditions, a daily mean direct aerosol forcing on surface global radiation between -8 and -64 W/m$^2$ for East Asia (e.g. Tab 1 in Kudo et al., 2010) and in the range -8 to -23 W/m$^2$ for Japan (in the Kanto region) has been reported (Kudo et al., 2010). However, under all sky conditions, it is generally thought that, depending on relative position/altitude of clouds and aerosol layer, aerosol effects on global solar radiation are usually small compared with cloud effects, while their impact on direct solar radiation is more important (e.g. Qu et al., 2017; Kosmopoulos et al., 2018). Therefore, under cloudless

conditions, AMATERASS estimates could be positively biased, depending on the actual aerosol load present at each location (Irie et al., 2017). Note that, in the following analysis, the potential overestimation of the satellite-based estimates caused by the absence of the aerosol correction will be investigated under clear sky conditions while the study concerning the RE effects above mentioned, which also affect the bias, will be performed under all sky conditions.

Although assimilation/forecast datasets from the European MACC project or the Japanese MRI/JMA are suitable sources of operational aerosol optical properties, aerosol estimations provided by Himawari-8 itself would be the best products to account for the aerosols effects in a future version of AMATERASS. Nevertheless, in order to implement such correction, two steps should be necessary. Firstly, the retrieval of aerosol optical properties (i.e. aerosol optical thickness and Angstrom exponent) must be accomplished by exploiting Himawari-8 observations. This activity has been initially performed by using

the algorithms of Higurashi and Nakajima (1999) for ocean and Fukuda et al. (2013) for land and recently further improved in the retrieval of urban aerosols (Hashimoto and Nakajima, 2017). Preliminary results are still under evaluation and initial validations with observations recorded at the Japanese SKYNET stations showed encouraging results. Therefore, the following step will be the inclusion of Himawari-8 aerosol parameters into EXAM and the creation of an updated version which will include the aerosol effects on the solar radiation. As shown in Takenaka et al. (2011), EXAM was designed to

account for aerosols in a neural network for clear sky conditions. In the original scheme, three aerosol optical properties (i.e. AOD, the imaginary part of the refractive index and the size distribution) and five additional parameters (i.e. solar zenith angle, surface albedo, surface pressure, ozone, water vapor) were included in the neural network and the achieved results were satisficing. A similar approach will be used in the next version of EXAM.

### 2.2 Surface observations

We carefully assessed the contribution of cloudiness and aerosols to SW satellite estimates at four stations in Japan (left panel of Figure 1) belonging to the ground-based SKYNET (http://atmos2.cr.chiba-u.jp/skynet/) network, i.e., Chiba (35.625°N, 140.104°E), Fukue-jima (32.752°N, 128.682°E), Cape Hedo (26.867°N, 128.248°E), and Miyako-jima (24.737°N, 125.327°E), using collocated measurements of surface radiation, aerosol properties, and total precipitable water

(PW) recorded between January and December 2016. Chiba is an urban site located near Tokyo, and Cape Hedo, Fukue-jima, and Miyako-jima are located on relatively small islands in the East China Sea. There are no major sources of air pollutants near the latter stations; however, they may be affected by aerosols from the desert and continental regions in East Asia.

The stations are equipped with pyranometers and radiometers to measure solar radiation and cloud and aerosol properties. We used a CM-21 pyranometer (Kipp & Zonen) to measure global solar (horizontal) irradiance from 285 to 2800 nm. This device was fully compliant with the highest ISO performance criteria and with the specifications for high-quality instruments as defined by the World Meteorological Organisation (WMO). It had a combined standard relative uncertainty of approximately 1.9% (Kratzenberg et al., 2006). The temporal resolution of this data set was 10 s.

Possibly, the reliability of the AMATERASS dataset needs to be tested under various climatological conditions. Therefore, ground-based measurements should include stations at high altitude, mainland, near the sea, near to urban sites etc. Chiba SKYNET station can be considered representative of urban conditions of the mainland region, while the other SKYNET stations can be roughly grouped as representative of a subtropical region possibly affected by desert and continental aerosols. Because of the necessity of examining other climates and different conditions and to extend our validation to the whole

Japan, we accomplished further comparisons with respect to 47 stations in the Japanese Meteorological Agency (JMA) surface network of pyranometers, some of them also belonging to the Baseline Surface Radiation Network (BSRN). The rigorous quality control of the measured data is regularly performed by JMA. This allowed to distinguish three main climatological regions and additional smaller areas presenting a distinct response.

For the SKYNET stations, aerosol optical properties, such as aerosol optical depth (AOD) and single-scattering albedo (SSA), were retrieved from direct sun and diffuse sky radiances measured using a sky radiometer (Model: POM-02; Manufacturer: Prede Co. Ltd., Japan). The core retrieval program was SKYRAD.pack (e.g., Nakajima et al., 1996). In response to recent worldwide SKYNET activity, we requested the single data set obtained in near real time using the common algorithm. A new Sky Radiometer analysis program package from the Center for Environmental Remote Sensing

(SR-CEReS, version 1, with skyrad.pack version 5; Mok et al., 2017) was subsequently developed and used in this study. We focussed on the AOD data set with a temporal resolution of 10 minutes.

Information on the total PW at the Chiba station was retrieved by microwave radiometer (MP1502). These measurements were compared to those of two other microwave radiometers (WVR-1125, MP1504) during an intensive campaign at Chiba University in November 2016 and proved to be of reasonable quality. The temporal resolution of this data set was 1 minute.

For the other stations, we used PW estimates from the ECMWF ERA-Interim data set.

**2.3 Clear-sky screening and clear-sky index**

Screening for clear-sky conditions, which is necessary to evaluate the influence of aerosols on solar radiation estimates, was conducted in a two-step process. We first selected only aerosol observations that were judged to be clear-sky observations by

the SKYNET algorithm (Khatri and Takamura, 2009). From among these observations, we retained radiation observations within a 10-min time window. Finally, we performed an additional screening using the standard deviation (SD) of the Himawari-based global solar irradiance data within a domain area of 20 km around each station. The SD mirrored the level of cloudiness within the considered area. We verified experimentally through a sensitivity analysis that an SD threshold of 10 W/m$^2$ was sufficient to ensure that data were recorded under clear-sky conditions. For selected days, we further verified the sky conditions by inspecting images taken using all-sky cameras.

Some of the following analyses were based on a computation of the so-called clear-sky index (CSI), here defined as the ratio of recorded irradiance to the corresponding expected clear-sky and aerosol-free irradiance simulated under the same conditions (i.e., in theory CSI should range from 0 to 1, although actual values sometimes exceed 1; see Section 3.1). In contrast to previous studies using empirical formulations (e.g., Piedehierro et al., 2014), we used an LUT-based approach to simulate clear-sky irradiance that used simulations run by the LibRadTran–UVSPEC library (Mayer et al., 2005). We used the DIScrete ORdinate Radiative Transfer (DISORT) solver (Stamnes et al., 1988) to solve the radiative transfer equations. DISORT was run under the usual conditions at each station and included observed PW (measured using a microwave radiometer at Chiba and retrieved by ECMWF ERA-interim at the other sites); additional parameters (e.g., ozone) remained fixed. In the following sections, in order to describe the sky conditions, we often refer to overcast, broken cloud, and clear-sky conditions for low, medium, and high CSI values, respectively. Although the CSI is a function of both cloud fraction and cloud optical depth (COD), using the clear sky index or clearness index for defining the three main categories of sky conditions is widely accepted (e.g. Serrano et al., 2006; Bech et al., 2015).

**2.4 Statistics and comparison**

The comparison of Himawari-8 estimates and ground-based SKYNET observations was performed at various time resolutions ranging from 2.5 minutes to 1 day. On the other hand, the additional comparison involving JMA measurements was made at time step of 10 minutes.

Consistent with previous validation studies (e.g., Antón et al., 2010; Nottrott and Kleissi, 2012; Damiani et al., 2012, 2013), the statistical comparison was performed in terms of the correlation coefficient (r), mean bias (MB), root mean square error (RMSE), and slope (a) of the regression line. In some of the analyses, we compared surface observations with respect to an average of satellite-based estimates taken within a variable spatial domain ranging from a single pixel (i.e., 1 km) to an area of 20 km around each SKYNET station. Unless otherwise noted, we focussed on data characterised by SZAs less than 70° to avoid possible cosine response error in the pyranometer and/or the satellite algorithm and the period from January to December 2016 (as a reference, at Chiba, under clear sky conditions, a global solar radiation of about 300 and 150 W/m$^2$ can be expected for SZA of 70° and 80°, respectively).

## 3 Results

### 3.1 Spatiotemporal variability of the satellite dataset

In this section, we briefly examine the spatiotemporal variability of the satellite data set compared to punctual ground-based observations (right panel of Figure 1). We exclude the SZA dependence of the irradiance to focus on the variability of the ground and satellite clear-sky indexes instead of the actual irradiance values. The variability is represented by the coefficient of variation $CV = (SD/mean) \times 100$ of the clear-sky index. We first compared the satellite instantaneous values (recorded every 2.5 min) with the corresponding surface observations. Then we evaluated the two data sets for different time step averages spanning from 5 min to 1 day (red to violet diamonds). Concerning the spatial domain, we started by comparing ground observations with the satellite pixel closest to the station (i.e., the distance from the station was 0 km; Figure 1); then we focussed on satellite estimates averaged over larger domains (i.e., within 5, 10, and 20 km of the station).

Figure 1 shows the difference between the CV of the surface SKYNET observations and that of the Himawari-8 estimates for different time and spatial domains as recorded at Chiba in August 2016. This difference was always positive, such that the variability of the surface observations was always greater than that of the satellite estimates. However, the difference between satellite and ground variability was smaller when data were averaged over long time domains (e.g., 1 h or 1 day) than over short time domains (e.g., 2.5, 5, or 10 min), whereas averaging over larger spatial domains reduced the variability of satellite-based estimates. These results confirm those of previous studies (e.g., Perez et al., 2016) and suggest that comparing satellite and ground observations is likely more challenging at short time scales than on a hourly or daily basis. Even the variability of single-satellite pixel results was smoother than the punctual ground variability, which indicates the small-scale variability of clouds at the subpixel level (i.e., <1 km). The variability became much less representative of that of the station within only a few (e.g., 5) kilometres. Finally, it is worth noting that the variability of Himawari-8 estimates became almost insensitive to the size of the domain at a daily time scale.

### 3.2 Validations under all-sky conditions

Because a solar radiation data set collected by the Himawari-7 (a previous Himawari) satellite with the EXAM algorithm had recently been validated using SKYNET observations (Khatri et al., 2015), we began by examining the results of our new validation for the same SKYNET stations at the same temporal resolution (i.e., a time step of 30 min; all available surface observations averaged within a time window of ±10 min; Figure 2). Thus, eventual improvements in the validation results could be more easily detected. Overall, the new analysis showed that the Himawari-8 estimates reproduced the observations at all stations very well, with statistics comparable to or better than those obtained by other geostationary satellites (Figure 2 and Table 1; cf. Gómez et al., 2016; Federico et al., 2017). For all stations, the correlation coefficient was approximately 0.95–0.96, and the (negative) MB was approximately 20–30 W/m$^2$ (i.e., Himawari-8 overestimated the ground-based observations) and the RMSE ranged from approximately 80 to 100 W/m$^2$. At all stations, the new data set yielded a lower

MB and RMSE than did the previous validation (Table 1). This small improvement over the results of the previous validation was perhaps due to an improved signal-to-noise ratio and/or spatial resolution in the new observations.

Although local surface features such as a complex morphology and/or surface albedo can affect the agreement between satellite estimates and ground-based measurements, cloudiness is expected to be the main factor determining the magnitude of RMSE. The left panel of Figure 3 shows the monthly RMSE between Himawari-8 estimates and ground-based JMA observations plotted on a map of the total cloud fraction in May 2016 retrieved from a Modern-Era Retrospective Analysis for Research and Applications (MERRA) reanalysis. Despite certain exceptions, a larger (smaller) RMSE is usually coupled with higher (lower) cloud fraction levels. This result is evident from the increasing gradient in the cloud fraction that developed from the South of Japan to the Okinawa Islands and that resulted in a corresponding increasing gradient in RMSE. Limited to the central and southern regions of Japan (enclosed by the black line), this overall influence of cloudiness on RMSE is also well evident in the scatter plot (inset panel).

The right panel of Fig. 3 shows the correlation between monthly RMSE and cloud fraction at each JMA station for January to December 2016 plotted over the Japan Digital Elevation Model derived from GTOPO-30 (https://lta.cr.usgs.gov/GTOPO30). Overall, the correlation was always positive but it ranged from more than 0.9 to less than 0.2 for the different stations. As Japan extends from north to south for about 3000 km, it is characterized by a variety of climatic regions which affected the pattern of the correlation. Indeed, according to previous studies (e.g. Ohtake et al., 2015), we can distinguish at least three main climatological regions and additional smaller areas. The first region (enclosed by the blue dashed line) is the norther part of Japan, mostly characterized by a subarctic climate, which provided a uniform response and small differences between Hokkaido and the north of the mainland (i.e. Tohoku). Here, except for two stations located in the Pacific sector of Japan, usually the correlation was lower than 0.4. Then, the large central region (within the red dashed line), which is characterized by humid and temperate climate, presents a more articulated pattern. The flat and strongly urbanized Kanto region (i.e. around Tokyo) showed the highest correlation values (r > 0.9). The SKYNET station of Chiba University is located here and it is supposed to be representative of this area. Then, correlations became slightly lower toward south with an evident distinction between the east coast, characterized by higher correlations, and the west coast, which presents lower values. It is worth noting that in winter the west coast is usually affected by elevate snowfall levels, while the Pacific coast usually shows frequent clear sky conditions. Fukue SKYNET station is located in the south-west sector of this region.

Stations in mountain regions usually present peculiar features that distinguish them from the stations located near the sea. A recent validation of three satellite-based radiation products over an extensive network of 313 pyranometers across Europe (Urraca et al., 2017) showed that stations sited in the Alps and Pyrenees have errors (i.e. RMSE and MB) two or three times larger than the ones of other locations. In mountainous regions, the altitude varies sharply and affects not only surface related parameters, but also the state of the atmosphere. Therefore, satellite models can fail in such areas because the spatial and

temporal resolutions are not high enough to account for the sharp terrain and changing weather conditions (Dürr et al., 2010; Castelli et al., 2014). Accordingly, although not shown here, in the central and norther Japan we found larger RMSE values for stations in mountainous regions compared with seaside stations. Finally, we note a somewhat uniform correlation (r > 0.6-0.8) in the subtropical region (enclosed by the green dashed line) along the Pacific Ocean where the largest precipitation

usually occurs. The SKYNET stations of Cape Hedo and Miyako are located in this latter region.

The peculiarities of the different regions are manifest when focusing on the annual cycle of RMSE (see inset in the right panel of Fig. 3). Generally, higher RMSE values were found in summer being the values highest in the subtropical region, followed by the central and the norther region. On the other hand, in winter RMSE values in the north were similar to the ones of the subtropical region. Future analyses, based on longer time series, are expected to further highlight the satellite

uncertainties at the different locations and, potentially, improve the accuracy of the satellite-derived dataset by site-adaptation methods (Polo et al., 2016).

Since reanalysis usually do not assimilate cloud fraction directly from observations, issues of MERRA in reproducing the cloud fraction for specific locations and periods can be expected. Moreover, uncertainties in the clouds identification from

15 the EXAM algorithm should be also taken into account. Both of them would affect the correlations shown in Fig. 3. We compared the monthly output of MERRA with visual observations recorded at the JMA stations and we found that, despite a systematic bias in the amount of cloudiness (reanalysis underestimates the cloud amount by about 0.1-0.2), MERRA satisfactory reproduced the month-to-month variability in 2016 in both central and north Japan (with some difficulty for the subtropical region). On the other hand, the EXAM algorithm can face difficulties in the cloud identification under bright

albedo (i.e. snow) conditions or in presence of small size clouds. For example, it is likely that the bright surface albedo would play a role in determining the different response in the east and west coast. Moreover, other factors such as the type of cloudiness or the location of the station, with possible constrains induced by the morphology and altitude, likely determined the RMSE.

Taking the above results into account, we excluded the stations of the norther region from the scatter plot (left inset of Fig. 3) and a good correlation (r = 0.85) between cloud fraction and RMSE for May 2016 was found. For points corresponding to stations in the subtropical region, we note that a large spread in RMSE was coupled to high cloud fraction (however, if these points are removed, the remaining ones still show an evident correlation (r = 0.74)). A somewhat small RMSE can also be expected, if a stable overcast sky dominates (Qu et al., 2017). Therefore, the larger spread could be related to the frequency

of such extreme conditions at the different stations of the subtropical region.

Determining effects due to the local morphology would require a detailed analysis beyond the objectives of this study. Figure 4 focuses on the influence of surface albedo on monthly mean differences (i.e., JMA measurements minus Himawari-8 estimates) in January 2016, when high (small as usual) albedo values characterise Japan at latitudes roughly north (south)

of Tokyo, likely because of snow on the ground. The retrieval of the surface albedo was based on a statistical approach which estimated the land surface albedo by the 2-nd minimum reflectance method over a 30 days time window (Fukuda et al., 2013). Although satellite estimates of shortwave radiation were generally larger than surface observations in snow-free locations characterised by low albedo values (around 0.1; i.e., in southern Japan), the opposite trend was observed in the

north (e.g., in Hokkaido) as well as in mountain regions (e.g., around Nagano). There the satellite estimates of solar radiation in winter tended to underestimate actual values by about 20 W/m$^2$ (see the scatter plot in the inset panel). As we will see in the next paragraphs, Himawari-based estimates of global radiation are usually larger than ground observations under cloudiness conditions, therefore this opposite trend appears somewhat anomalous. As an example, the bottom-right inset shows the diurnal variation of the radiation under high surface albedo conditions at Asahikawa (in Hokkaido, see the pink

arrow) on January 21. Ground JMA observations suggest that while the morning was generally characterized by small cloudiness and radiation values close to the average of the season, in the afternoon a sudden and intense reduction of the radiation, likely caused by thick clouds, occurred. In contrast, Himawari-based estimates are extremely low in the morning while much larger than ground observations in the afternoon. Previous studies showed that the cloud identification from passive satellite instruments is usually affected by large uncertainty under bright albedo conditions (e.g., Chan & Comiso,

2013; Damiani et al., 2015) and this can cause biased estimates of surface solar radiation (Tanskanen et al., 2007). Indeed, a possible explanation of the anomalous AMATERASS estimates could rely on a potential underestimation of the surface albedo from the EXAM algorithm which leads to misinterpretation of the observed bright scene as clouds. However, more detailed analysis, beyond the objective of the present work, will be necessary to univocally identify the issue and, eventually, improve the algorithm.

As mentioned in the Introduction, under broken cloud conditions the surface radiation can be even greater than the corresponding radiation under an ideal cloud-free sky (i.e., CSI >1). This phenomenon is called radiation enhancement (RE), and it is likely to occur when the sky is partially covered by thick clouds while the solar disk remains cloud free (Gu et al., 2001; Piedehierro et al., 2014) or when the solar disk is partially or fully obstructed by optically thin clouds (Gueymard,

2017). Previous studies have shown that RE up to around 50% can be observed and simulated with two-dimensional radiative transfer models (Piedehierro et al., 2014; Pecenak et al., 2016). The left panel of Figure 5 shows the global solar radiation measured at the Chiba station on 6 August, 2016, as well as the simulated radiation expected under clear-sky conditions. An RE event lasting for more than 40 min occurred around noon, and its flux was approximately 100 W/m$^2$ larger than the simulated value (see the pattern of the sky recorded by the all-sky camera around the peak of the RE). Recall

that these enhancements cannot be reproduced by current satellite-based algorithms. Figure 5 also shows the frequency distribution of RE events at the four SKYNET stations examined in this study. A strong seasonality in RE occurrence is apparent, with the largest frequency of events (about 10–20% of all observations) concentrated in July–September, with a lower frequency (usually <5%) during the other seasons. At Chiba, the mean (additional) flux caused by REs in August was 91 W/m$^2$ and the standard deviation was 48 W/m$^2$. Note that although REs are nearly equally distributed during the day, the

occurrence of the strongest events was dependent on SZA. Nevertheless, because solar radiation was low under such conditions, these events did not strongly affect the amount of radiation at the ground; we therefore did not include them in our analysis (we considered only data with SZA <70°). Previous studies (Piedehierro et al., 2014) found that at middle/low SZA (roughly < 40°) the occurrence of REs is almost equally distributed while, for larger SZA (roughly > 40°), their

frequency decreases along with increasing SZA. On the other hand, simulations showed that for the overhead zenith angle the magnitude of the REs is highest for low COD (about 2-3), while their magnitude increases with increasing SZA and reach the maximum for high COD values (Pecenak et al., 2016). According to these previous results, our dataset showed that, although REs were nearly equally distributed during the day, the occurrence of the strongest events occurred mostly at high SZA. We also checked the relation between COD and REs by exploiting Himawari-8 COD data (Nakajima &

Nakajima, 1995) for Chiba station. In the greatest majority of the cases, REs resulted to be coupled with COD of about 0.5-5, while only few of them were associated with higher COD.

RE events are not typically expected to play a main role in the global solar radiation budget. For example, at Chiba they accounted for only 1.55% of the global radiation budget in August. Nevertheless, to exemplify the large variability in the

15 time length and magnitude of REs, the right column of Figure 5 shows a record-long RE event recorded at Chajnantor station (Chile, 5100 m a.s.l.), belonging to the ESR European Skynet network, under comparable (austral) summer conditions (i.e., on 24 January, 2017). Details on the state-of-the-art measurements of shortwave radiation recorded there, on their comparison with other stations and radiative transfer simulations as well as details on the influence of the altitude on the solar irradiance can be found in Cordero et al. (2014, 2016). In this location, a single RE episode, with flux approaching 300

W/m$^2$ greater than the simulated clear-sky flux, developed between 8:00 and 11.30 A.M. and was characterised by an average irradiance greater than the solar constant. Additional spectral observations of global irradiance made around the peak of the RE event were compared to analogous measurements recorded under clear-sky conditions on 29 January, 2017. The largest enhancement (up to about 250 W/m$^2$/nm) occurring around 450 nm was somewhat lower than that of the global irradiance, as was expected given the low time resolution of these spectral observations (Gueymard, 2017). The time length

and magnitude of this episode suggest that large RE effects can sometimes occur; therefore, it would be desirable for satellite algorithms to take them into account.

The left column of Figure 6 compares the distribution of the CSI (between 0 and 1, we did not include cases where CSI >1) of the ground-based SKYNET observations and Himawari-8 estimates at different stations. Note that the distributions are

30 shown at a (satellite) instantaneous temporal resolution of 2.5 min, collocated with the averages of all available surface observations within a time window of ±1.25 min. The Himawari-based CSI is shown at the pixel level (red line) and as the average of the 5 km domain (black line). At the pixel level, the satellite estimates seem to better reproduce the distribution of ground observations, although there was an overestimation in clear-sky cases and an underestimation in cases with substantial cloudiness (CSI ca. 0.2–0.5), particularly at the southernmost stations (i.e., Cape Hedo and Miyako), whereas the

number of overcast cases (CSI ca. 0.1) was usually well reproduced. When considering a somewhat larger spatial domain, the distribution remained almost similar to that of the mid-range CSI values (ca. 0.2–0.6), although there was a reduction in the size of the peaks for extreme cases (i.e., clear-sky and overcast conditions). Further expanding the spatial domain to 10 and 20 km changed the shape of the distribution only slightly; therefore, we did not include these data in Figure 6.

The insets in the right column of Figure 6 show the distribution of the ground-based CSI for the RE events (i.e., CSI >1). To take into account the combined uncertainties of the pyranometer and simulation, we plotted only REs with an absolute difference between observation and simulation of greater than 5%. The distribution of the RE events in the different locations was quite similar; however, more cases of extreme CSI (i.e., >1.2) occurred at Chiba than at other stations. Although the satellite algorithm could not reproduce the REs, it is interesting to examine what the satellite sees under such

conditions. This is shown in the main panels of the right column: At the pixel level, the large majority of events were interpreted by the EXAM algorithm as characterised by clear-sky conditions (i.e., CSI ca. 0.9–1). These cases ranged from 55% to 75% of the total, with Miyako showing the largest proportion. The satellite estimates averaged within the 5 km domain showed a similar distribution, peaking under clear-sky conditions, but with a more gradual decrease toward more noticeably cloudy conditions.

Figure 7 shows a scatter plot of the instantaneous global SW radiation at a time step of 2.5 min between Himawari-8 estimates and ground-based observations at the four SKYNET stations. We attempted to classify the cloudiness in six ranges based on previously computed ground-based clear-sky index (0–0.3, 0.3–0.6, 0.6–0.9, 0.9–1, 1–1.1, and RE events as defined in the previous section), and we applied this classification to the same scatter plots by highlighting the different CSI

ranges with different colours. Then, we calculated statistics for each range (Figure 7) and for the overall data set (Table 2). We confirmed using the all-sky cameras that overcast conditions were mainly associated with CSI <0.3, whereas clear-sky conditions were mainly associated with $0.9 < CSI < 1.1$. Broken clouds were associated with intermediate values. As shown in Table 2, a somewhat good agreement between satellite and ground-based data was found, although, as already shown in Figure 2, Himawari-8 tended to slightly overestimate the actual surface radiation. This overestimation was larger for low

irradiance values but tended to decrease for higher values and even show an opposite trend (underestimation) under large/extreme irradiance conditions. The latter case corresponds to the occurrence of RE events (violet dots), as reflected in the somewhat large RMSE observed at all stations. In general, the statistics were slightly worse than those for the previous comparison (Figure 2), as expected given the higher temporal resolution of the data sets used in the new analysis. The correlation between the two data sets improved with a decrease in cloudiness (as CSI increased). By contrast, the mean bias

(SKYNET minus Himawari-8) was consistently negative for the supposed overcast conditions and tended to decrease toward clear-sky conditions.

Note that this general pattern did not depend on the size of the region within which the Himawari's pixels were averaged (e.g., there were virtually no differences between areas extending to 10 or 20 km) but was mainly due to heterogeneity in the cloudiness within the satellite pixel when data were taken at nearly instantaneous time steps. Hourly and daily statistics are

also shown in Table 2. As expected, results based on hourly and daily averages show an improved correlation and reduced RMSE with respect to the results based on the instantaneous data. By contrast, the mean bias results were only slightly improved.

Because RE events are not expected to be reproduced by the EXAM algorithm, it is interesting to examine comparison statistics excluding these events. These additional results are reported in Table 2 in brackets. Excluding REs slightly improved the correlation coefficient and RMSE between the two data sets for an instantaneous time step. By contrast, at longer time steps, although the correlation did not show any substantial change, the RMSE showed an opposite trend, i.e., the RMSE became somewhat worse at a time step of 1 h and greatly increased at a daily time scale with changes of about 10–30% over values computed including RE events. By contrast, the mean bias became about 10 W/m$^2$ larger than that previously computed; this effect did not depend on the time step. This RE impact was likely underestimated given the conservative threshold of 5%, which reduced the number of identified RE events. We suggest that researchers should take RE events into account, particularly when conducting satellite validation in cloudy regions.

The first two upper panels of Figure 8 (I and II) show diurnal variation in the difference (i.e., SKYNET minus Himawari-8) in seasonal global radiation for spring, fall, winter, and summer at the four SKYNET stations. Absolute (relative) differences show a satellite overestimation in the range of 20–80 W/m$^2$ (5–15%). It is interesting that the absolute difference did not peak in summer, as one would expect given the higher irradiance flux, but in spring. Nevertheless, summer differences became much larger if RE events were not included. This trend was evident at all stations. Note that, by comparing the SUNY modelled dataset with weather stations in California, a considerable larger bias in the diurnal variation (annual mean around 18 % and up to about 50 % in summer) was reported by Nottrott and Kleissl (2010).

The two lower panels (III and IV) show the CSI diurnal variation. The interseasonal variability of the ground-based index was very well captured by the satellite-based index for all stations. Even the diurnal variability, which generally showed somewhat less cloudiness around noon than in early morning or late afternoon, was very well reproduced. Although winter was the season with the least cloudiness at Chiba, it was the cloudiest season at the other stations, where summer was least cloudy.

### 3.3 Validations under clear-sky conditions

Both aerosols and PW strongly reduced the amount of radiation that reached the ground. Although work is ongoing to include aerosol forcing in the EXAM algorithm, the current version of the AMATERASS data set does not take aerosols into account while including PW as estimated by the Japanese reanalysis. Figure 9 shows the diurnal variation in global irradiance measured at the ground and estimated by satellite at the Chiba station on 2 days (25/01/2016 and 21/05/2016) mainly characterised by clear-sky conditions. Although in May Himawari-8 clearly overestimated the actual radiation at the surface, in January it tended to slightly underestimate it. Moreover, 21 May was affected by a high aerosol load, with AOD

peaking well above 0.6; 25 January presented very low aerosol amounts, with AOD below 0.05. This clearly suggests direct forcing by aerosols on global surface radiation.

In the following analysis, we focus only on clear-sky data. As mentioned in Section 2, the SD of the satellite measurements within the area surrounding our station provided information on cloudiness. Therefore, we used an SD threshold of 10 W/m$^2$ together with the method of Khatri and Takamura (2009) to remove data affected by cloudiness. We further collocated the instantaneous radiation observations collected under clear-sky conditions with the aerosol property measurements. The top panels of Figure 10 show scatter plots of the difference between ground SKYNET and Himawari-8 global radiation data sets with respect to the AOD during the period January to December 2016 at the Chiba and Fukue stations. The linear regression line clearly shows an increasing satellite overestimation of solar radiation along with the increasing aerosol load. The overall instantaneous direct radiative forcing efficiency estimated by the slope of the linear regression was about –125 W/m$^2$ per AOD unit at the Chiba station. The direct radiative forcing efficiency of the AOD was slightly larger at Fukue (about –146 W/m$^2$). This result is consistent with those of previous studies showing that Fukue is usually characterised by thicker aerosols (Dim et al., 2013). Overall, these results are roughly in agreement with findings reported in modelling studies (Irie et al., 2017) and previous observational results (e.g., Xia et al., 2007; Huttunen et al., 2014). Indeed, computations performed by a radiative transfer model under the usual conditions at Chiba showed that the direct radiative forcing efficiency at SZA of 50° is about –150 W/m$^2$ per AOD unit (Irie et al., 2017). A similar value (–146.3 W/m$^2$) was experimentally estimated from observations recorded by a CIMEL radiometer at Liaozhong (China) (Xia et al., 2007) and comparable estimates were highlighted for other AERONET stations around the world (Huttunen et al., 2014) as well as for the Mediterranean basin (e.g. Cachorro et al., 2008; Di Biagio et al., 2009, 2010; Papadimas et al., 2012). These studies reported a direct radiative forcing efficiency at the surface of about 100-150 W/m$^2$ (Cachorro et al., 2008), 100-200 W/m$^2$ (Di Biagio et al., 2009, 2010) and 100 W/m$^2$ (Papadimas et al., 2012) per AOD unit. Recall that these values are based on the instantaneous difference between ground observations and satellite estimates (which do not take aerosol effects into account) under clear-sky conditions. We verified that variation in the PW explains approximately 50% of the AOD variance at Chiba; therefore, water vapour could potentially create confusion and mask these results if there are inconsistencies between the actual PW and the forecast values used in the satellite algorithm. However, we also computed the forcing efficiency using usual clear-sky simulations including PW measurements retrieved by the microwave radiometer and verified that, as expected, differences in the results were not substantial.

It is worth noting that the instantaneous forcing efficiency depends also on airmass (i.e. SZA). In a recent paper (Irie et al., 2017), by running radiative transfer simulations including aerosol proprieties observed at the SKYNET Chiba station, we showed that the relative decrease of solar radiation induced by an increasing AOD is larger at high SZAs. In some previous studies based on surface measurements (e.g. Xia et al., 2007; Cachorro et al., 2008; Di Biagio et al., 2009, 2010), the instantaneous forcing efficiency has been computed for small SZA ranges as a preliminary step to estimate the mean direct aerosol forcing. However, usually small differences in the forcing efficiency have been reported for SZA of 10-60°. Thus,

because of the additional constrains due to the satellite acquisition times and clear sky observations and since data recorded at high SZAs were previously excluded, in Fig. 10 we showed the forcing efficiency for the whole dataset. An additional issue is due to the fact that the instantaneous forcing efficiency can be only partially described by a simple linear fit and, for the same aerosol family, the estimation of the change in solar radiation for AOD unit is rather different if computed over

different AOD ranges. In our dataset, while we have a large range of AOD for mid SZAs, we have a much smaller range for high SZAs which is probably insufficient to extrapolate a reliable efficiency. Moreover, the slope of the linear regression is also affected by the different type of aerosols (e.g. Di Biagio et al., 2009, 2010), therefore sampling the data for different SZA could introduce further issues if episodes of different aerosol types occurred along the year. In the bottom panels of Fig. 10 we tried to remove the SZA effect on AOD by computing the percentage decrease of solar radiation in function of the

aerosol optical depth slant (AODS i.e. AODS = AOD/cos(SZA); e.g. Garcia et al., 2006) and we obtained -14.3 and -14.5 % per AODS unit at Chiba and Fukue, respectively.

To highlight the impact of aerosol load on downwelling global radiation on a seasonal time scale, we plotted the daily cycle of mean radiation differences (Figure 11) between SKYNET and Himawari-8 and the AOD for spring, fall, and winter (not

enough clear-sky measurements were available in summer). Overall, at the Chiba station, the largest negative differences between the ground SKYNET observations and Himawari-8 estimates occurred in spring (about 20–30 W/m$^2$), followed by fall (<10 W/m$^2$), whereas the AMATERASS estimation matched the observations in winter. These differences reflect the larger AOD amounts in spring, followed by fall and winter. These results are roughly consistent with a previous investigation (Kudo et al., 2010), based on ground-based observations carried out in a close location (Tsukuba city, distance

about 50 km), which reported a direct aerosol forcing larger for spring to summer (-22 W/m$^2$) and small in winter (-8.3 W/m$^2$).

Although the seasonality of the aerosol load distribution appeared to be somewhat different at Fukue, the negative differences in spring were close to those in Chiba and were coupled with a similar AOD amount. There was a peak in the impact of aerosols in winter, with a reduction of the global radiation up to about 60 W/m$^2$. Finally, the minimum levels of

AOD in fall were coupled with anomalous positive differences, particularly in the early morning and afternoon. Overall, it is noteworthy that the difference between SKYNET and Himawari-8 irradiances was anti-correlated with AOD for spring and winter, when AOD variations were relatively large. This finding suggests the importance of the impact of aerosol load on global radiation from the viewpoint of their diurnal variation.

A recent study showed that the diurnal variation patterns in Himawari-8 AOD data were consistent with those seen in

SKYNET observations (Irie et al., 2017). Thus, Himawari-8 aerosol products would provide a unique spatial and diurnal variation information which, once included in the next version of the EXAM algorithm, is expected to reduce the positive bias under clear sky conditions here highlighted.

Since the different weather conditions finally determine the validation, it is not straightforward to compare our findings with validation results obtained in previous studies dealing with other satellites/algorithms. Nevertheless, our results appear to be at least comparable to the ones previously achieved. Since there is a lack of concurrent operational products of surface solar radiation with a resolution comparable to AMATERASS (i.e. at 2.5 min), in the introduction we focused on results obtained from datasets with a resolution of 1 h and 15 min. We noted that the majority of the previous studies (e.g. Nottrott and Kleissi, 2010; Thomas et al., 2016a,b; Qu et al., 2017) reported a general overestimation of the estimated solar radiation by a few tens of W/m$^2$. This occurred despite the fact that aerosols were taken into account in the algorithm. A comparable overestimation, which did not depend on the adopted temporal resolution, has been found in our analysis. This suggests that usually the role of aerosols could be marginal compared with the influence of cloudiness when dealing with all-sky conditions. In particular, for stations located in the flat and strongly urbanized Kanto region (e.g. Chiba station), the all-sky overestimation ranged from a few percent in winter to about 10 % in summer (with a MB of about 23 W/m$^2$ on annual basis) and roughly followed the annual cycle of the cloud amount (i.e. smaller in winter and larger in summer). On the other hand, the RMSE between observations and satellite-based estimates did depend on the resolution (see Tab. 2) and, in our analysis, it was reduced to about one third of its original value (computed from data at 2.5 min) when statistics were computed from daily means.

We also showed that the RMSE computed from high resolution (i.e. 10 min) estimates usually ranged between 60 and 90 W/m$^2$ with larger values occurring in summer, especially in the subtropical region (see the inset in the right panel of Fig. 3). Such values resulted to be comparable to RMSE values computed at similar resolution for Europe (i.e. 15 min, Qu et al., 2017) or even to values from hourly datasets for Canada and Italy (Djebbar et al., 2012; Federico et al., 2017). In the latter study (Federico et al., 2017) the authors reported a RMSE in the range of 150-200 W/m$^2$ for three mountain stations (two of them over 2000 m a.s.l.). It is worth noting that such values are higher than any RMSE obtained for Japan from the dataset at 2.5 min, even for subtropical stations. The authors suggested that in these locations RMSEs were affected by specific constrains induced by the morphology and/or the larger amount of cloudy days with respect to other locations. Among the SKYNET and JMA stations in Japan not one is located at such high altitudes so unfortunately we cannot compare this interesting feature. However, since we found the largest RMSE values for the subtropical islands, we checked if they were associated with less clear sky observations. By using the CSI between 0.9 and 1 as a proxy of clear sky conditions, it resulted 5 % more clear sky observations at Chiba than in the other subtropical SKYNET locations.

## 4 Conclusion

This study evaluated the AMATERASS global solar radiation data set based on Himawari-8 observations and the EXAM algorithm with respect to ground truthing observations and focussed on spatiotemporal variability effects (mainly induced by cloudiness) and the aerosol load. In general, the EXAM algorithm applied to Himawari-8 performed well in reproducing the

surface global irradiance at the four SKYNET stations examined in this study. The MB was in the range of 20–30 W/m$^2$ (i.e., EXAM overestimated ground observation), whereas the RMSE was usually around 80 W/m$^2$ (slightly larger at Miyako).

Comparisons with respect to the JMA stations showed that the magnitude of the RMSE was mainly determined by the level of cloudiness during the period under investigation. By contrast, bright albedo conditions led to a reduction or even a reversal of the sign of the mean differences between ground observations and Himawari-based estimates.

The agreement with ground-based observations depended on the time step used in the validation exercise as well as on the spatial domain and climatological regions. Worse agreement was found for the instantaneous time step (2.5 min), with the best RMSE at the daily level (Table 2). In particular, the RMSE depended heavily on the time step used, whereas the MB remained roughly constant.

At the pixel level, a larger number of samples were interpreted as clear-sky data by the EXAM algorithm than in the real atmosphere. This trend was only slightly reduced when a larger spatial domain was considered. Overall, AMATERASS tended to slightly overestimate actual surface radiation. Moreover, the overestimation was larger under overcast conditions, whereas frequent episodes of surface REs (i.e., measured radiation larger than the expected clear-sky radiation) under broken cloud conditions tended to compensate the bias.

The influence of the RE events appears to be substantial, particularly in summer, when they accounted for 10–20% of the total measurements. The EXAM algorithm could not reproduce such events and interpreted these situations as characterised by almost clear-sky conditions. Obviously, AMATERASS underestimated the global radiation during periods affected by REs; this tendency balanced the general overestimation associated with cloudiness. Indeed, removing RE events from the comparison increased the MB at the instantaneous time step. This finding was also evident at longer time steps. By contrast, the RMSE was reduced at the instantaneous time step, as one would expect given the reduced number of spikes, whereas it increased at the daily scale.

Under clear-sky conditions, the influence of aerosols makes the AMATERASS estimates larger than ground observations. Based on the SKYNET and AMATERASS data sets in 2016, the overall instantaneous direct radiative forcing efficiencies were about –125 and –146 W/m$^2$ per AOD unit at Chiba and Fukue, respectively. We found that the diurnal pattern of the difference between SKYNET observations and AMATERASS estimates in irradiance at these stations was anti-correlated with AOD in different seasons, with larger (smaller) differences in periods of higher (lower) AOD. At Chiba, the maximum differences occurred in spring (about 20–30 W/m$^2$), whereas at Fukue in winter they were greater than 40–50 W/m$^2$. The impact of aerosol load on global radiation from the viewpoint of their diurnal variation was also discussed.

Overall, our analysis confirmed the good accuracy of the AMATERASS solar global radiation product at temporal resolution of 1 h and 10 min but showed larger deviations at 2.5 min. An improved algorithm better accounting for the cloud effects would certainly alleviate such deviations which are more evident under fast varying gradients induced by changing clouds. For example, AMATERASS "saw" slightly more clear sky scenes compared to surface observations while it tended to overestimate the solar radiation under cloudiness conditions. A finer satellite spatial resolution would improve the identification of small cumulus clouds. Moreover, both REs and aerosols contributed to these deviations but their impact was

much smaller. Nevertheless, the portion of these deviations, which arises from the intrinsic difference in comparing "punctual" measurements and satellite pixels, could be hardly removed. When the pixel is partly cloudy, surface measurements can greatly vary depending on whether clouds are located or not along the solar beam path to the sensor possibly resulting in a satellite overestimation or underestimation of the irradiance at ground. Due to the large field of view

5 of the pyranometer and depending on the solar zenith angle conditions, even clouds located at kilometers from the station can potentially affect observations. Unless such effects are accounted by a sufficiently long temporal integration of the surface observations, this would result in discrepancies with respect to satellite-based estimates.

Despite these issues, we expect AMATERASS to play a key role in contributing to the development of an efficient EMS in Japan.

### Acknowledgments

The present study was supported by JST/CREST/EMS/TEEDDA fund, Grant Number JPMJCR15K4, Japan.

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

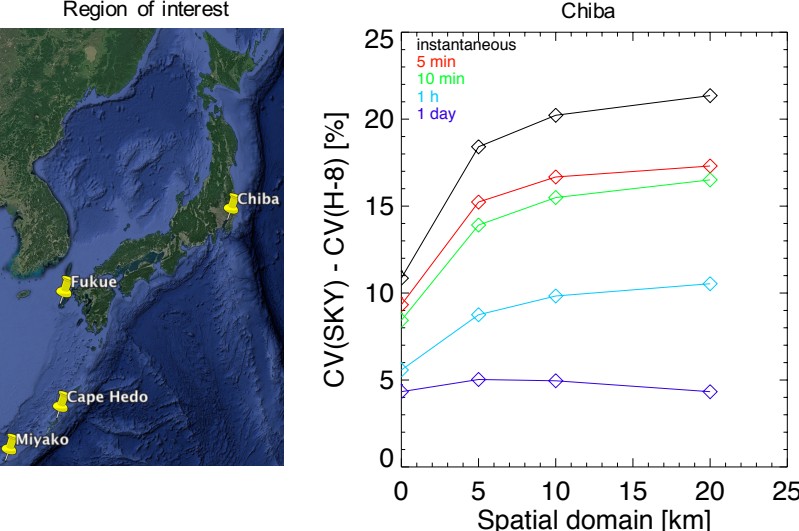

**Figure 1: (Left panel) Region of interest showing the four SKYNET stations used in the validation of Himawari-8 estimates of global radiation. Source: Google Earth Pro. (Right panel) Difference in the coefficient of variation (CV) of global radiation values obtained by SKYNET (SKY) and Himawari-8 (H-8) and in the size of the domain for different time averages (instantaneous time resolution: 2.5 min). Period: August 2016; station: Chiba. See text for further details.**

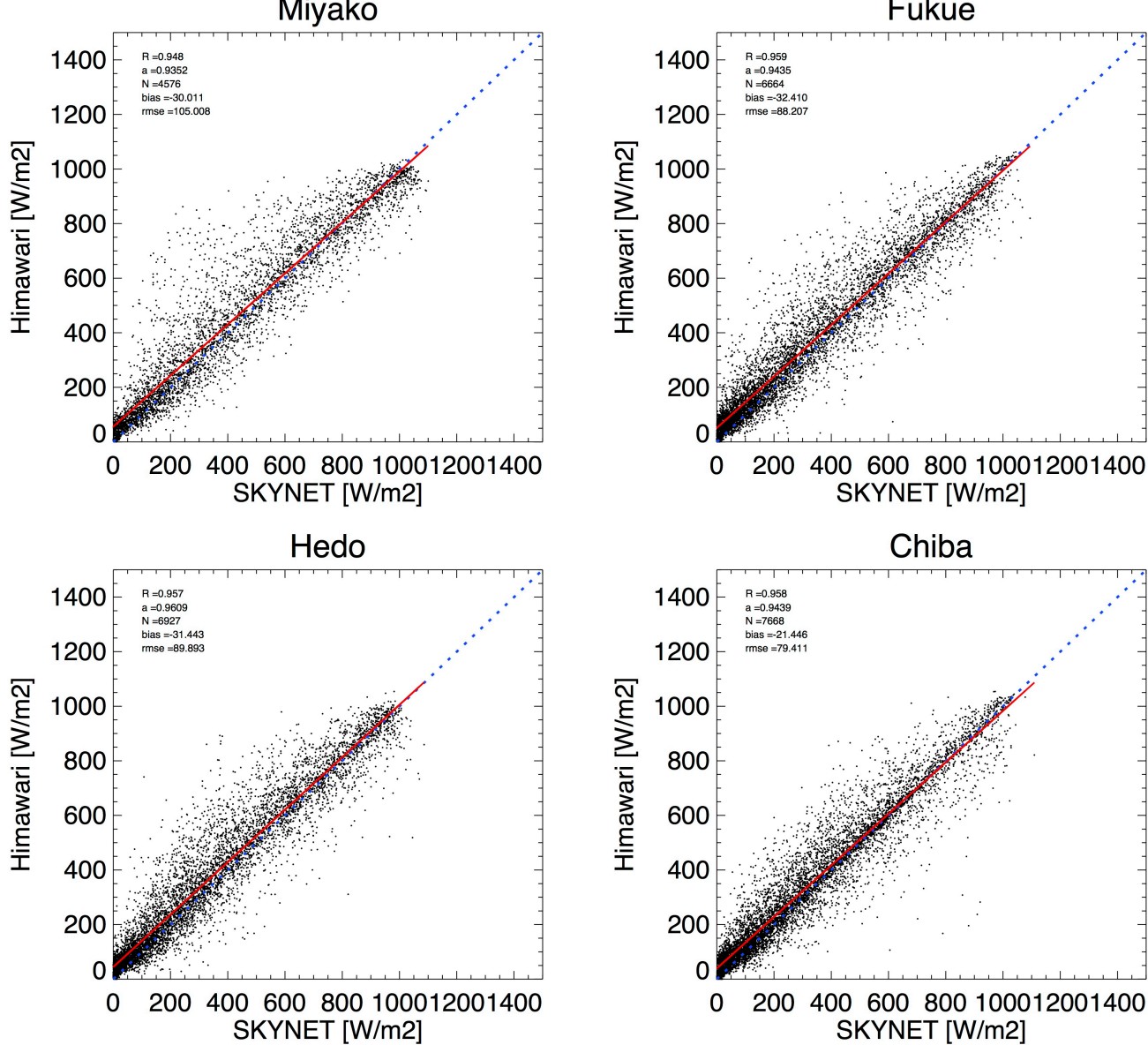

**Figure 2: Scatter plot of ground-based SKYNET observations and satellite-based estimates of surface global irradiance from Himawari-8 at the Chiba, Fukue, Miyako, and Cape Hedo stations in 2016. Statistics describing the comparison, i.e., correlation coefficient (r), slope of the regression line (a), number of samples (N), mean bias (bias), and root mean square error (RMSE), are shown in the upper left corner; the dashed line is a 1:1 line, and the regression line is shown in red. Time step: 30 min.**

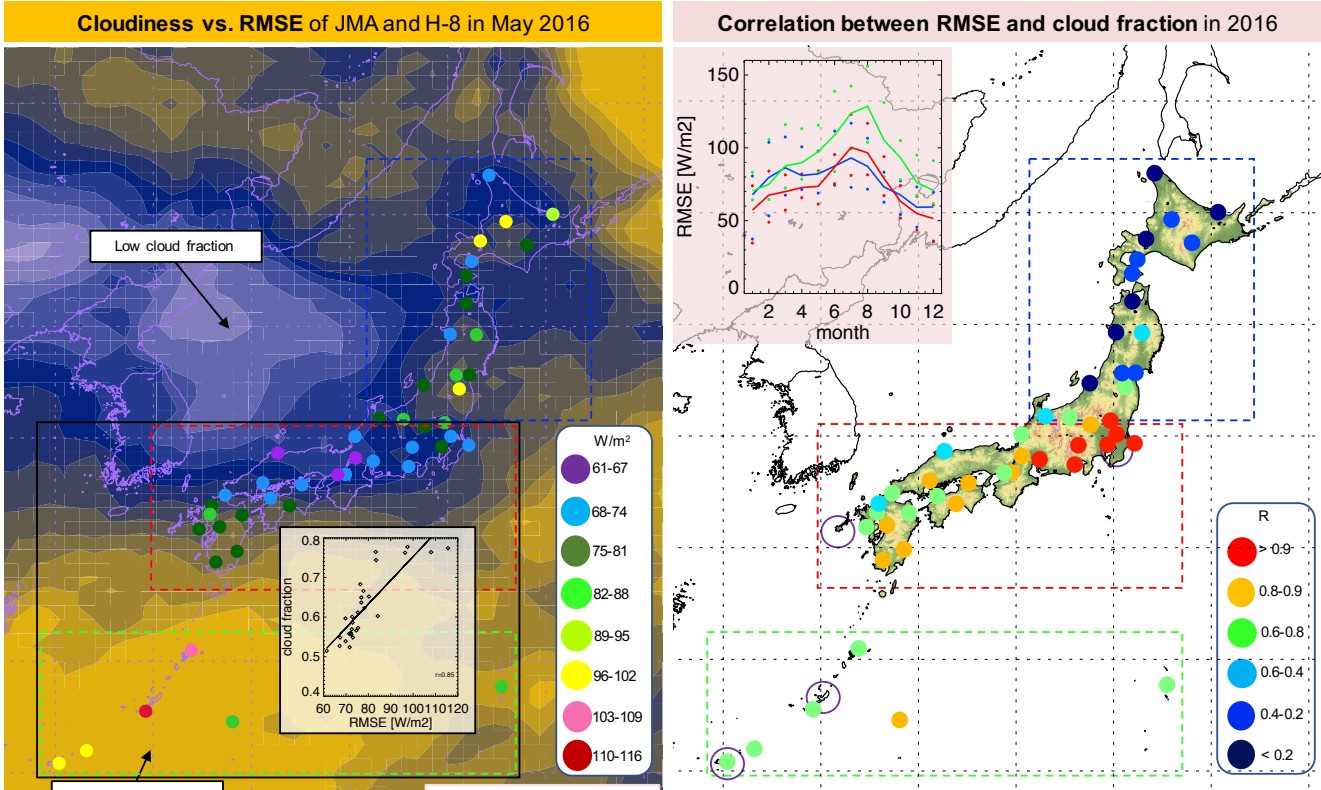

**Figure 3: (Left panel) Influence of cloudiness (i.e., total cloud fraction, violet to yellow contour map) on the monthly RMSE of ground observations and Himawari-8 estimates of solar radiation at the 47 stations (points) in the Japanese Meteorological Agency (JMA) network in May 2016. Bottom inset: Scatter plot of the total cloud fraction and RMSE for the stations in the central and south regions (i.e. within the area delimited by the black line). (Right panel) Correlation between monthly RMSE and cloud fraction at each JMA station for 2016 plotted over the Japan Digital Elevation Model derived from GTOPO-30 (https://lta.cr.usgs.gov/GTOPO30). The three main regions, i.e. north, central and south, are enclosed by blue, red and green dashed lines, respectively; violet circles show the location of the SKYNET stations; inset: mean RMSE (bold lines) and minimum and maximum RMSE (points) for the different regions for December to January 2016.**

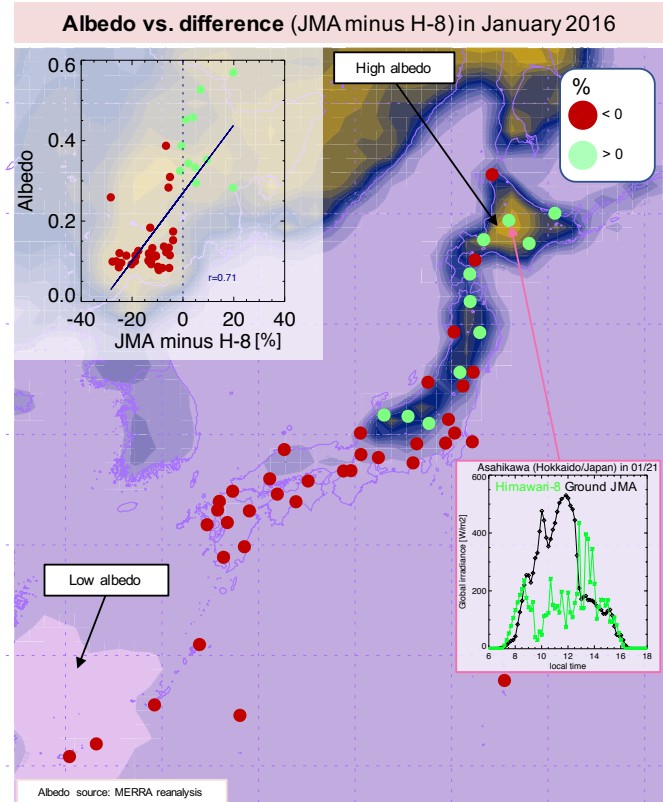

**Figure 4: Influence of surface albedo (violet to yellow contour map) on the monthly mean difference between ground observations and Himawari-8 estimates at the stations (points) of the JMA network in January 2016. Top-left inset: Scatter plot of albedo and difference values for all stations. In the main panel and the inset negative (positive) differences are shown as red (green) dots. Bottom-right inset: ground-based JMA observations (black line) and Himawari-8 estimates (green line) at Asahikawa (Hokkaido/Japan, see the pink arrow) on January 21 under very high albedo conditions. The original time step of the comparisons is 10 minutes.**

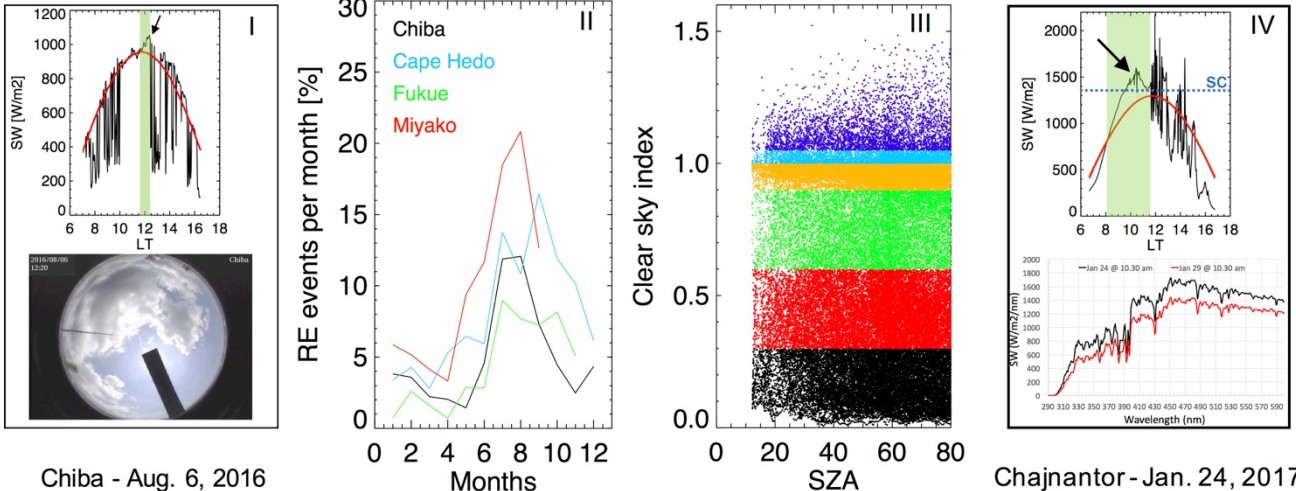

Chiba - Aug. 6, 2016                                    Chajnantor - Jan. 24, 2017

**Figure 5: From left to right panels: (I) Downwelling global shortwave radiation (SW) measurements (black line) and clear-sky simulation (red line) at the Chiba station on 6 August, 2016 (top panel). An example of a radiation enhancement (RE) event lasting about 40 min (green shadow) is indicated by the black arrow and also shown in the all-sky camera image below around the peak of the event (bottom). (II) Percentage of RE events per month among total monthly samples for the different SKYNET stations. (III) Clear-sky index (CSI) and solar zenith angle (SZA) at the Chiba station. (IV) Global SW measurements (black line) and clear-sky simulation (red line) at the Chajnantor station (Chile) on 24 January, 2017. The arrow indicates an RE event lasting about 3.50 h (green shadow); the horizontal dotted line indicates the solar constant (top panel); spectral SW measurements recorded at the Chajnantor station at 10.30 A.M. on January 24 (i.e., around the peak of the RE event; black line) and under clear-sky conditions on 29 January, 2017 (red line; bottom panel).**

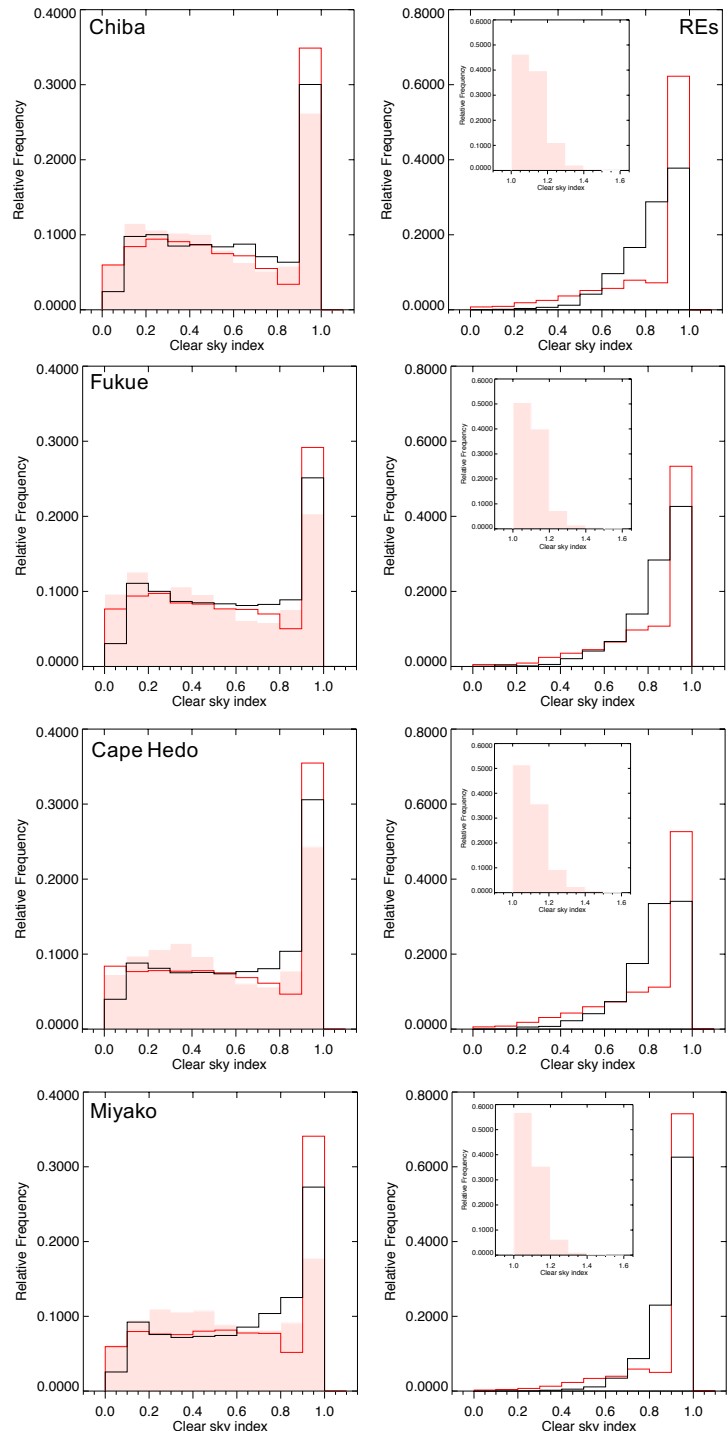

**Figure 6: (Left column) Distribution of the CSI based on ground-based SKYNET observations (pink area) and Himawari-8 estimates at the pixel level (red line) and within a 5 km spatial domain (black line) for the different SKYNET stations. (Right**

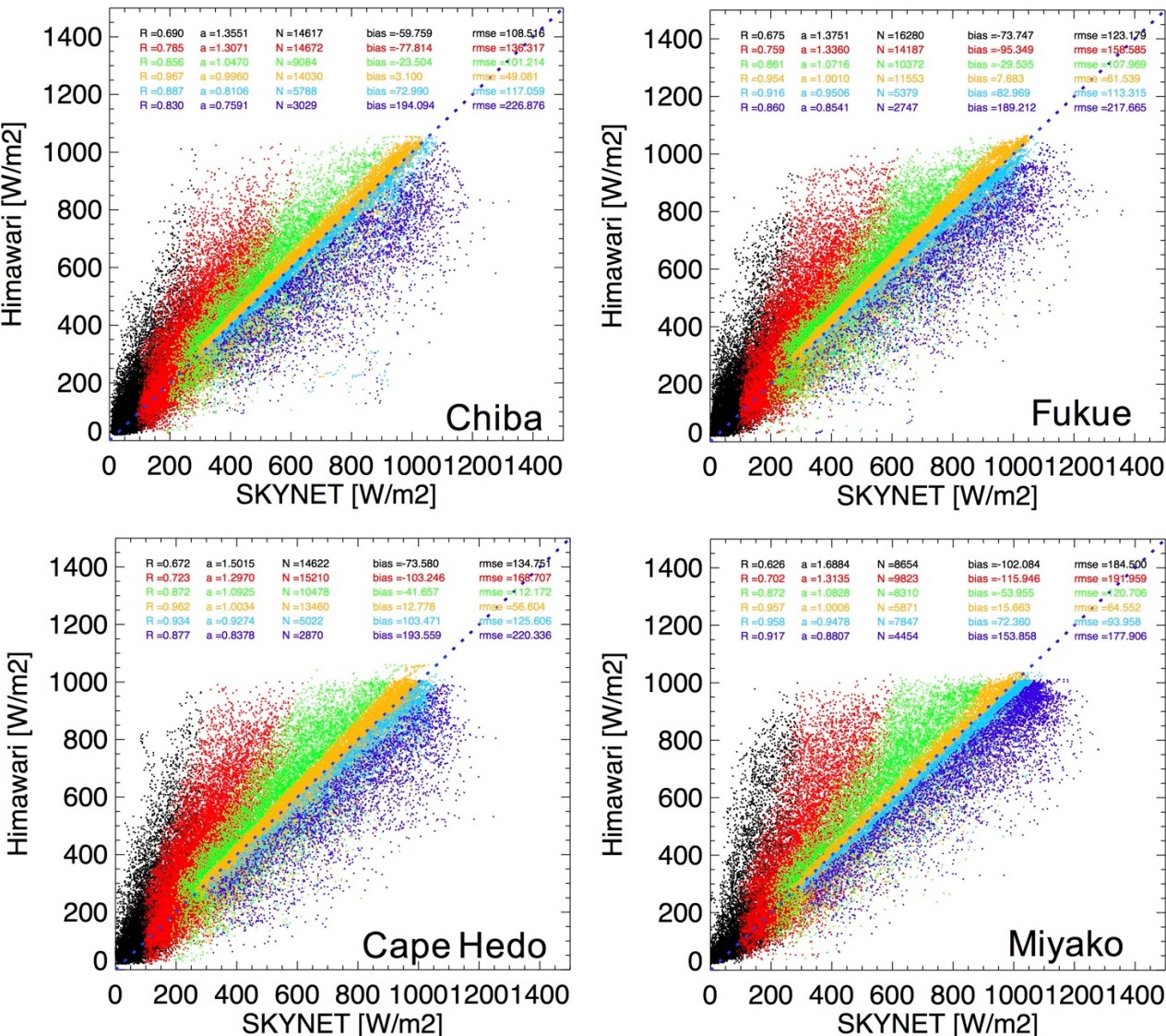

column) Main panel: distribution of the CSI based on Himawari-8 estimates as in the left column but only for RE events. Inset: distribution of the CSI based on SKYNET observations but only for RE events. Time step: 2.5 min. See text for further details.

5 **Figure 7: Scatter plot of surface global irradiance data obtained by ground-based SKYNET observations and Himawari-8 estimates at the four SKYNET stations. Colours indicate different clear-sky index ranges (based on SKYNET observations): black (0–0.3), red (0.3–0.6), green (0.6–0.9), orange (0.9–1), light blue (1–1.1), and violet (RE events). Statistics describing the comparison, i.e., correlation coefficient (R), slope of the regression line (a), number of samples (N), mean bias (bias), and root mean square error (RMSE), are shown; the dashed line is a 1:1 line. Time step: 2.5 min. See text for further details.**

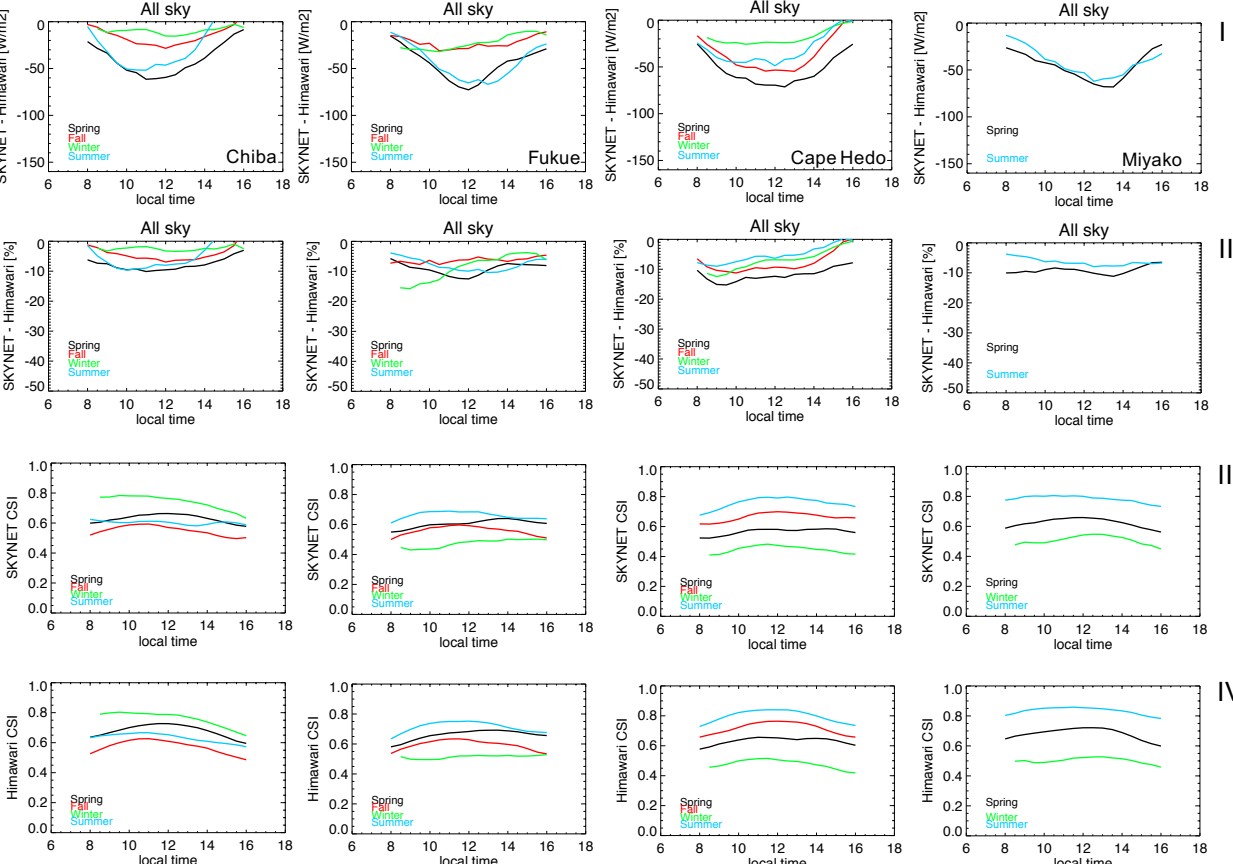

**Figure 8: Rows I and II: Diurnal variation in the difference (i.e., SKYNET data minus Himawari data) in seasonal global radiation for spring, fall, winter, and summer (top: absolute differences, bottom: relative differences). Rows III and IV: Diurnal variation in seasonal CSI for spring, fall, winter, and summer (top: SKYNET-based CSI, bottom: Himawari-based CSI). A running average of 1 h is applied to the instantaneous data. Results are shown for (left to right) Chiba, Fukue, Cape Hedo, and Miyako.**

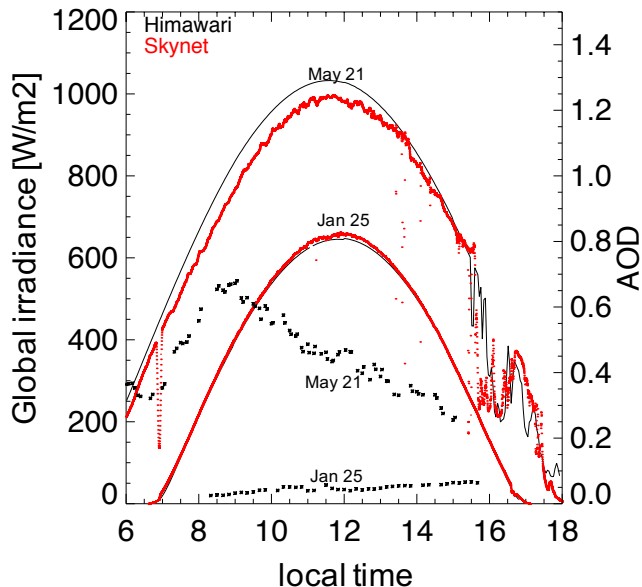

**Figure 9: Global surface irradiance measured at the ground (red line) and estimated by Himawari-8 (black line) on 2 days (25/01/2016 and 21/05/2016) mainly characterised by clear-sky conditions at the Chiba station. Measured SKYNET aerosol optical depth (AOD) values are also shown for both days (black points, see opposite axis).**

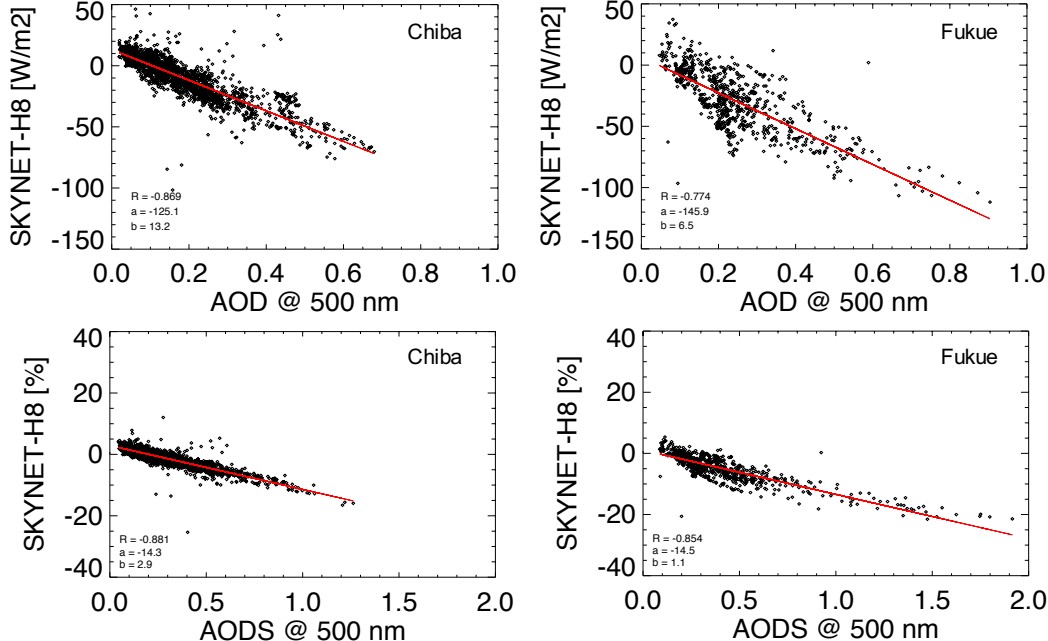

**Figure 10: (top panels) Scatter plots of the difference in clear-sky surface global radiation (SKYNET observations minus Himawari-8 estimates) and measured SKYNET AOD at 500 nm under clear-sky conditions between January and December 2016 at Chiba (left) and Fukue (right) station. The regression line (in red), its slope (a), intercept (b) and the correlation coefficient (R) are also shown. (bottom panels) Scatter plots as above but for percentage difference in global radiation and aerosol optical depth slant (AODS).**

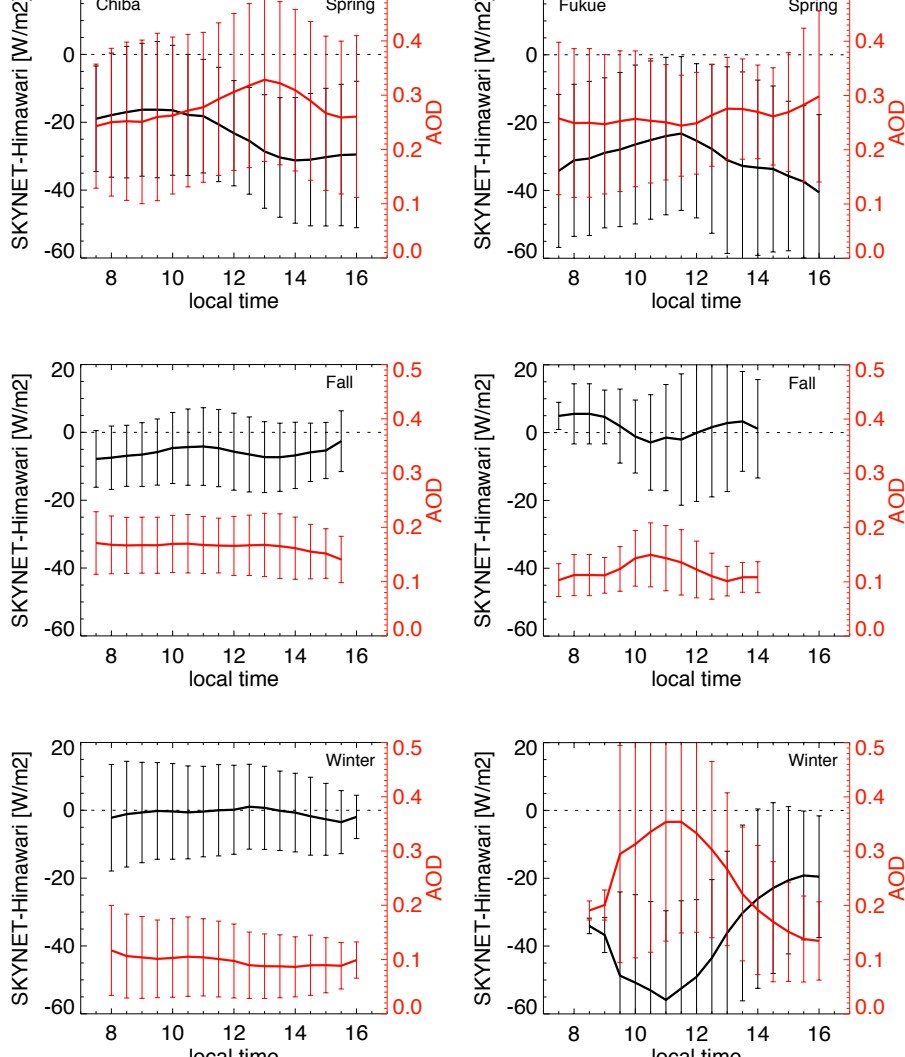

**Figure 11: Diurnal variation in the seasonal global radiation difference (i.e., SKYNET data minus Himawari-8 data, black lines) under clear-sky conditions and AOD (red lines on the opposite axis) for (top to down) spring, fall, and winter at the Chiba (left column) and Fukue (right column) stations. Error bar shows the standard deviation.**

| Satellite / year | Himawari-8 / 2016 | | | | Himawari-7 / 2015 | | | |
|---|---|---|---|---|---|---|---|---|
| Station | Chiba | Fukue | Hedo | Miyako | Chiba | Fukue | Hedo | Miyako |
| Time step | ½ h | ½ h | ½ h | ½ h | ½ h | ½ h | ½ h | ½ h |
| N | 7668 | 6664 | 6927 | 4576 | 7508 | 6742 | 7390 | 5563 |
| r | 0.958 | 0.959 | 0.961 | 0.948 | 0.959 | 0.952 | 0.956 | 0.913 |
| MB (W/m2) | -21.446 | -32.410 | -31.443 | -30.011 | -28.974 | -44.445 | -43.229 | -36.725 |
| RMSE (W/m2) | 79.411 | 88.207 | 89.893 | 105.008 | 83.933 | 97.744 | 97.369 | 131.332 |
| a | 0.9439 | 0.9435 | 0.9609 | 0.9352 | 0.970 | 0.982 | 0.986 | 0.905 |

**Table 1 - Comparison of Himawari-8 and Himawari-7 data sets with surface SKYNET observations at the Chiba, Fukue, Cape Hedo, and Miyako stations in 2016 (see Figure 2) and 2015, respectively. Results of the statistical analysis (N = number of samples, r = correlation coefficient, MB = mean bias, RMSE = root mean square error, a = slope of the regression line) are reported for a ½-h time step.**

| Time step | 2.5 min | | | | 1 h | | | | 1 d | | | |
|---|---|---|---|---|---|---|---|---|---|---|---|---|
| Station | Chiba | Fukue | Hedo | Miyako | Chiba | Fukue | Hedo | Miyako | Chiba | Fukue | Hedo | Miyako |
| N | 59816 (56787) | 59128 (56381) | 60185 (57315) | 42426 (37972) | 2796 (2790) | 2687 (2686) | 2782 (2779) | 1953 (1951) | 315 (315) | 288 (288) | 309 (309) | 198 (198) |
| r | 0.909 (0.925) | 0.910 (0.921) | 0.897 (0.909) | 0.879 (0.885) | 0.961 (0.962) | 0.961 (0.961) | 0.956 (0.962) | 0.948 (0.945) | 0.977 (0.975) | 0.976 (0.977) | 0.979 (0.980) | 0.966 (0.956) |
| MB (W/m2) | -22.902 (-34.477) | -33.798 (-44.663) | -34.043 (-45.439) | -33.256 (-55.204) | -20.450 (-31.460) | -32.478 (-41.865) | -31.998 (-42.223) | -31.392 (-51.675) | -20.068 (-30.858) | -31.425 (-41.421) | -31.927 (-42.850) | -31.371 (-52.527) |
| RMSE (W/m2) | 113.317 (103.827) | 125.544 (119.251) | 132.283 (126.270) | 151.303 (147.870) | 70.741 (73.085) | 81.935 (84.611) | 81.579 (83.829) | 95.545 (104.758) | 44.638 (51.152) | 55.729 (60.770) | 52.024 (59.072) | 60.775 (77.852) |
| a | 0.8498 (0.9030) | 0.8497 (0.8913) | 0.8422 (0.8820) | 0.8000 (0.8592) | 0.9528 (0.9818) | 0.9463 (0.961) | 0.9679 (0.9900) | 0.9327 (0.9763) | 0.9613 (0.9812) | 0.9550 (0.9762) | 1.0003 (1.0248) | 0.9456 (0.9846) |

**Table 2 - Comparison of the Himawari-8 data set with surface SKYNET observations at the four SKYNET stations. Results of the statistical analysis (N, r, MB, and a) are reported for time steps of 2.5 min, 1 h, and 1 day. Values in brackets indicate the same data sets excluding radiation enhancement (RE) events.**