# Peer review of "Evaluation of Himawari-8 surface downwelling solar radiation by ground-based measurements"

_Atmospheric Measurement Techniques, 2017_

## Referee Comment (RC1) · Anonymous Referee #1 · 25 Jan 2018

This study evaluates the AMATERASS product of the Himawari-8 satellite based on the EXAM algorithm. Ground-based measurements from SKYNET, JMA and BSRN were used for this evaluation under all-sky and clear sky conditions. It is a well written paper which with some minor revisions it could be published in the AMT journal.

My most serious comments have to do (i) with the lack of comparable results from the bibliography and (ii) with the non-inclusion of an aerosol input to the model. This fact makes the evaluated product almost blind to the aerosol effects. However, the impact of this additional uncertainty was assessed, but there is still a reliability gap. I recommend the authors at least to provide some information about potential sources of operational aerosol optical properties and ways of integrating them into the EXAM algorithm (additional extension or a new model approach?). On the other hand, the

discussion about RE effects is valuable for the improvement of satellite algorithms as to take them into account, but the overall impact in terms of estimated solar energy potential for the natural energy resource exploitation is meaningless compared to the aerosol effect.

In the Introduction section, the authors provide good bibliography about hourly-based validations but there is no reference for higher temporal resolution. There is a need here for additional references using satellite data in finer time steps.

On page 2, lines 3-4, rephrase as "However, to implement such an EMS system, surface solar irradiance data must be supplied as accurately as possible."

On page 2, lines 29-30, the percentages seem to refer to relative RMSE.

On page 3, line 7, provide reference.

Page 3, lines 20-21: Indeed the RE can introduce bias into the evaluation of satellite estimates. Mention other uncertainties (altitude corrections, aerosol optical properties (aod, angstrom, ssa), sza and shading from adjacent mountains) with relevant references.

Page 4, line 10: The EXAM algorithm will include the aerosol effect in the same neural network or in a separate one only for cloudless conditions? The combination of clouds and aerosols in the same NN could result further uncertainties with aerosol mixtures, aerosol and cloud mixtures, multiple aerosol and cloud layers etc. Explain in brief how this inclusion will be addressed.

Page 4, line 11: The mentioned bias from the absence of aerosol inputs may be confused with the aforementioned RE and this is an issue for the reliability and accuracy of results. A clarification is needed here (e.g. underestimation for RE conditions and overestimation because of aerosols). Provide also reference for the aerosol impact on radiation and relevant sensitivity analysis in order to quantify the overall bias by the absence of aerosol input.

Page 4, line 20: The applicability of EXAM with the Himawari-8 input, needs to be tested under various climatological conditions, so the selection of ground-based measurements may include stations in high altitude, mainland, near the sea, near to urban sites like Chiba etc. Here, one station is near urban site, and three affected by desert and continental regions. Discuss the representativeness of the stations selected as well as further necessary test cases for future similar evaluations.

The overall evaluation was performed with Skynet, BSRN and other stations from the JMA, so the Title of this paper could be optionally renamed as "... by ground-based measurements".

On page 5, line 25 and page 9, line 30, is there a classification of sky conditions in the bibliography based on the CSI? This will be helpful for the readers in order to have a sense of quantification for the various sky conditions.

Page 6, line 4: Mention additionally the minimum solar energy potential in such large SZAs for strengthening this consideration.

On page 7, lines 4-5, there is a need for discussion of these results (MB and RMSE) with the mentioned (line 3) or additonal references, as to provide direct comparison with similar approaches and satellites.

Page 7, line 17: This is an important aspect and it needs a short description of the magnitude of this effect (with reference).

Page 7, line 22: Is there a need for an altitude correction for the satellite estimations?

Page 10, line 24: Provide comparable results from the bibliography.

Page 11, line 25: What about high aerosol loads without the impact of PW? Huttunen et al. (2014) explains only this (effect of water vapor on the determination of aerosol direct radiative effect). A more focused reference is needed. Percentages of energy attenuation are also welcome (lines 22-23) in order to be useful for pv reduced production because of high aerosol loads.

Finally, on page 12, lines 4-5, provide comparable results from the bibliography. In the end of this paragraph the author may provide a short description of the potential EXAM upgrade with the inclusion of the aerosol impact.

The high spatial and temporal resolution of the Himawari-8 satellite in conjunction with near real-time algorithms like the EXAM, will improve the precision of the solar farms planning and production control with clear benefits for the local energy transmission and distribution system operators. This paper after the above corrections could be a step forward to the efficient and full integration of the natural energy resource to the electricity grid and will contribute to the development of upgraded energy management systems.

---

## Referee Comment (RC2) · Anonymous Referee #2 · 26 Jan 2018

Review of the manuscript "Evaluation of Himawari-8 surface downwelling solar radiation by SKYNET observations" by Damiani et al., 2018.

Summary: The manuscript describes the validation of the solar irradiance dataset from the Himawari-8 satellite by surface based observations using the SKYNET and JMA networks in Japan for the whole year 2016. The solar irradiance dataproduct from Himawari-8 is obtained from the EXAM algorithm. The comparison is performed between surface point observations and satellite derived solar irradiance data ed for different spatial and temporal resolutions. Several influencing parameters such as aerosol optical depth and surface albedo is investigated. The largest part of the paper discusses the quality of the solar irradiance product with respect to varying cloudiness, which is responsible for the largest variation in solar irradiance. The overall objective

is to validate solar irradiance retrievals at high temporal and spatial resolution in order to use these datasets in near realtime for informing photovoltaic installations of the expected solar power because rapid transients can be harmful to PV plants, as stated in the manuscript at page 3, line 3.

Comments: The manuscript is well written and has a good structure. The results and conclusions are discussed and presented quite objectively so that the reader can make his own opinion as to the quality of the comparison. The validation of the dataset over only one year of measurements is rather short and might not allow to draw conclusions as to the long-term performance of the satellite-derived solar irradiance product. Nevertheless the results are encouraging and worth to be published as they provide a very good view of the current state-of-the-art in retrieving surface solar radiation from an assimilation of surface based and space-based data on an unprecedented time and spatial resolution.

In contrast to the authors I conclude from this study that the solar radiation product as presented here is far from reaching the objectives stated in the introduction and which seem to be the rationale for the satellite itself. In general I have the impression that the authors have drawn very optimistic conclusions from their dataset. I suggest modifying the abstract and conclusions to provie a more objective assessment of the satellite solar irradiance product, especially in the frame of the overall objective of using the high temporal solar data for PV applications.

- The all-sky solar radiation data shows large deviations with respect to the surface based datasets, especially at the high temporal resolution of 2.5 minutes. This is very well seen in Figure 6, where deviations between the SKYNET pyranometer and Himawari-8 are strongly dependent on cloudiness (Clear Sky index is used here) with deviations which can exceed a factor of 3! With regard to the stated objective to use these high temporal resolution data for near realtime forecast for PV applications, especially under fast varying gradients induced by changing clouds, I would like to have this aspect more clearly addressed in the discussion and conclusion section of the

manuscript. Specific comments:

-Page 5, line 25: The CSI is also a function of the cloud optical depth, not only cloud fraction. A fully overcast sky with cirrus clouds might have a CSI close to 1. Does cloud type have a significant influence on the dataproduct?

-page 6, line 10. I assume this should be irradiance, instead of radiance?

- page 7, lines 13-16 and Figure 3, left inset: I disagree with the statement made by the authors that the data supports the assumption that a larger RMSE is correlated with cloud fraction. The left inset to Figure 3 is made up of two distinct dataclouds, 6 at high cloud fraction, of which 4 are outliers, and only the two leftmost which confirm the authors's assumption. If these 6 datapoints at high cloud fraction are neglected, then the remaining datapoints show no correlation at all between RMSE and cloud fraction. The linear fit is too suggestive and not representative for the dataset.

- page 7, discussion on surface albedo. I have understood that the surface albedo is retrieved from the satellite data itself, and that problems occur when clouds and albedo are misclassified. My question: Is the surface albedo retrieved independently for every time slot, e.g. every 2.5 minutes, or does the algorithm use the fact that while clouds can very extremely fast, surface albedo will be slowly varying, on the scale of days or more?

-page 8, line 17-18. It would the interesting to know more about the dependency between RE and SZA and add some quantitative values.

-page 11 and Figure 9. In my opinion the solar radiation forcing with respect to AOD should depend to some extent on airmass, e.g. SZA, since the attenuation of solar radiation is mainly from the direct beam radiation and thus follows the Beer-Lambert law.

- Conclusion, page 13, line 14. I do not see how this statement is an outcome of this paper, since the only comparison to an other satellite product was Himawari-7.

Figures: Figure 2: The resolution is not very good and the values shown in the subsets are not easy to read.

Figure 3: As mentioned in my previous comment, the suggestive linear fit between RMSE and cloud fraction is not convincing, since it depends on the two outlier points at high RMSE and high cloud fraction. Another view could be that there the top six points are outliers, and the remaining points show no clear correlation between RMSE and cloud fraction.

Figure 6: The text in the four sub figures is very difficult to read, possibly for lack f resolution, and the chice of colors. Especially the yellow text is unreadable. The variables need to be defined in the caption.

Figure 9: as mentioned previously, I expect the aerosol radiative forcing to also depend on airmass. Is it possible to add this information, or mention it in the accompanying text?

Figure 10: I suggest to plot the right (red) AOD axis in reverse, to show even more clearly the anti-correlation between AOD and residuals. In the caption, the error bars need to be better explained. Does it for example represent the standard deviation?

Table 2: I assume that the information given in table 2 should be consistent with the information given in Figure 6? In that case, it would be better to harmonise this information, and give the slope in the same units, not the inverse.

---

## Author Response (AR1)

Dear Editor,

We are sending back the revised version of our manuscript where we addressed all comments raised from reviewers.
We are grateful to Referee #1 and Referee #2 for carefully reading our manuscript and for their helpful suggestions that allowed improving the quality of the study. Here we first report the referee comments (in black), and then we provide our responses (in blue). In our replies, pages and lines (p. xx, l. xx) refer to the updated manuscript.

Best regards,

Alessandro Damiani

List of relevant changes made in the manuscript

1 Inclusion of additional validation results from the bibliography and discussion of our findings in the light of these previous results
2 Plans for future improvements of the EXAM algorithm
3 Discussion on the representativeness of the selected stations within the various climatological conditions of Japan (updated Fig. 3)
4 Discussion on the quality of the validation results at high temporal resolution (2.5 min)
5 Inclusion of a more extensive analysis of the correlation between RMSE and cloud fraction (updated Fig. 3)
6 Discussion on the dependence of the aerosol forcing on airmass i.e. SZA (updated Fig. 10)

Anonymous Referee #1

This study evaluates the AMATERASS product of the Himawari-8 satellite based on the EXAM algorithm. Ground-based measurements from SKYNET, JMA and BSRN were used for this evaluation under all-sky and clear sky conditions. It is a well written paper which with some minor revisions it could be published in the AMT journal.
R -> We are grateful to Referee #1 for carefully reading our manuscript and for the helpful suggestions that allowed improving the quality of the study.

My most serious comments have to do (i) with the lack of comparable results from the bibliography and (ii) with the non-inclusion of an aerosol input to the model. This fact makes the evaluated product almost blind to the aerosol effects. However, the impact of this additional uncertainty was assessed, but there is still a reliability gap. I recommend the authors at least to provide some information about potential sources of operational aerosol optical properties and ways of integrating them into the EXAM algorithm (additional extension or a new model approach?). On the other hand, the discussion about RE effects is valuable for the improvement of satellite algorithms as to take them into account, but the overall impact in terms of estimated solar energy potential for the natural energy resource exploitation is meaningless compared to the aerosol effect.
R -> Concerning point (i):
Due to limitations in the temporal resolution of the satellite products on which the algorithms are based, there is a lack of concurrent operational products of surface solar radiation with a resolution comparable to AMATERASS (i.e. at 2.5 min). Therefore, in the introduction of the revised manuscript (p. 3, l. 5-16), we reviewed additional references of validations performed at 15 min (Kosmopoulos et al., 2018; Qu et al., 2017; Ruf et al., 2016; Zo et al., 2016). Then, following your comment below (cf. page 7, lines 4-5), in the revised discussion section (p. 17, l. 1-27), we compared our findings with such previous results (both at 15 min and 1 h).

Concerning point (ii):
Although assimilation/forecast datasets from the European MACC project or the Japanese MRI/JMA are suitable sources of operational aerosol optical properties, aerosol estimations provided by Himawari-8 itself would be the best products to account for the aerosols effects in a future version of AMATERASS. Nevertheless, in order to implement such correction, two steps should be necessary. Firstly, the retrieval of aerosol optical properties (i.e. aerosol optical thickness and Angstrom exponent) must be accomplished by exploiting Himawari-8 observations. This activity has been initially performed by using the algorithms of Higurashi and Nakajima (1999) for ocean and Fukuda et al. (2013) for land and recently further improved in the retrieval of urban aerosols (Hashimoto and Nakajima, 2017). Such preliminary results are still under evaluation and an initial validation with observations recorded at the Japanese SKYNET stations showed encouraging results. Then, the following step will be the inclusion of Himawari-8 aerosol parameters into EXAM and the creation of an updated version which will include the aerosol effects on the solar radiation. As shown in Takenaka et al. (2011), EXAM was designed to account for aerosols in a neural network for clear sky conditions. In the original scheme, three aerosol optical properties (i.e. AOD, the imaginary part of the refractive index and the size distribution) and five additional parameters (i.e. solar zenith angle, surface albedo, surface pressure, ozone, water vapor) were included in the neural network and the achieved results were satisficing. A similar approach will be used in the next version of EXAM. In the revised manuscript, we included the above discussion at p. 5, l. 15-28.

In the Introduction section, the authors provide good bibliography about hourly-based validations but there is no reference for higher temporal resolution. There is a need here for additional references using satellite data in finer time steps.

R -> In the revised introduction (p. 3, l. 5-16) we included further references of validations performed at high temporal resolution and described their main results (e.g. Ruf et al., 2016; Zo et al., 2016; Qu et al., 2017; Kosmopoulos et al., 2018). Then, following your comment below (cf. page 7, lines 4-5), in the updated discussion section we discussed our findings in the light of these previous results (p. 17, l. 1-27).

On page 2, lines 3-4, rephrase as "However, to implement such an EMS system, surface solar irradiance data must be supplied as accurately as possible."

R -> Thank you!

On page 2, lines 29-30, the percentages seem to refer to relative RMSE.

R -> Yes, thank you!

On page 3, line 7, provide reference.

R -> We included a further reference (Perez et al., 2016)

Page 3, lines 20-21: Indeed the RE can introduce bias into the evaluation of satellite estimates. Mention other uncertainties (altitude corrections, aerosol optical properties (aod, angstrom, ssa), sza and shading from adjacent mountains) with relevant references.

R -> in the revised manuscript (p.4, l.4-13), we included an additional discussion as follows: "Many additional factors can introduce a bias in the results of a validation exercise and make satellite-based estimates more uncertain (Polo et al., 2016). Among others, we remind the important role played by aerosols in reducing the surface solar irradiance, thus this is an important parameter to be accounted in the algorithms especially for polluted regions or deserts under clear sky conditions (e.g. Qu et al., 2017). Then, it is worth to mention the negative effect of the complex morphology of mountainous regions which cannot be easily accounted due to the limited satellite spatial resolution and the local fast-changing weather conditions (Dürr et al., 2010; Urraca et al., 2017; Federico et al., 2017). Further, the negative impact of the high solar zenith angle (SZA) and satellite viewing zenith angle on the quality of the estimates must be also mentioned: the former can cause low clouds to be overshadowed by higher clouds while the latter produces a parallax effect in the clouds position (Polo et al., 2016; Qu et al., 2017). Finally, uncertainties in the surface albedo especially over bright surfaces (e.g. snow or desert) would make hard cloud identification (e.g. Tanskanen et al., 2007)."

Page 4, line 10: The EXAM algorithm will include the aerosol effect in the same neural network or in a separate one only for cloudless conditions? The combination of clouds and aerosols in the same NN could result further uncertainties with aerosol mixtures, aerosol and cloud mixtures, multiple aerosol and cloud layers etc. Explain in brief how this inclusion will be addressed.

R -> In the revised version of the manuscript, here (p. 5, l. 15-28) we included the discussion at point (ii) described above. In particular, EXAM was originally designed to account for aerosols in a neural network for clear sky conditions (Takenaka et al., 2011). In the original scheme, three aerosol optical properties (i.e. AOD, the imaginary part of the refractive index and the size distribution) and five additional parameters (i.e. solar zenith angle, surface albedo, surface pressure, ozone, water vapor) were included in the neural network and the achieved results were satisficing. A similar approach will be used in the next version of EXAM.

Page 4, line 11: The mentioned bias from the absence of aerosol inputs may be confused with the aforementioned RE and this is an issue for the reliability and accuracy of results. A clarification is needed here (e.g. underestimation for RE conditions and overestimation because of aerosols).

R -> In the revised version of the manuscript (p. 5, l. 11-13), we further discussed and made clear this potential issue. The overestimation of the satellite-based estimates caused by the absence of the aerosol correction has been evaluated only under clear sky conditions. On the other hand, the investigation concerning the RE effects, which contribute to decrease the positive bias of the satellite-based estimates, has been conducted under all sky conditions.

Provide also reference for the aerosol impact on radiation and relevant sensitivity analysis in order to quantify the overall bias by the absence of aerosol input.

R -> In the revised version of the manuscript here (p. 5, l. 3-10) we included further references for the aerosol impact on radiation (e.g. Xia et al., 2007; Cachorro et al., 2008; Di Biagio et al., 2009, 2010; Papadimas et al., 2012; Huttunen et al., 2014) and we reported that, under clear sky conditions, for East Asia previous studies found a daily mean direct aerosol forcing on surface global radiation ranging from -8 to -64 W/m$^2$ while for Japan (in the Kanto region) roughly in the range -8 to -23 W/m$^2$ (Kudo et al., 2010).

Moreover, we stated that although aerosols modulate the amount of solar radiation reaching the ground under all sky conditions, it is generally thought that, depending on relative position/altitude of clouds and aerosol layer, usually aerosol effects are small compared with cloud effects for the solar global radiation while their impact is more important for the direct solar radiation (e.g. Qu et al., 2017; Kosmopoulos et al., 2018).

Page 4, line 20: The applicability of EXAM with the Himawari-8 input, needs to be tested under various climatological conditions, so the selection of ground-based measurements may include stations in high altitude, mainland, near the sea, near to urban sites like Chiba etc. Here, one station is near urban site, and three affected by desert and continental regions. Discuss the representativeness of the stations selected as well as further necessary test cases for future similar evaluations.

R -> Following your comment, in the revised manuscript (p. 6, l. 10-18) we further extended the validation results and discussed the necessity of an additional analysis for specific climatological conditions. Here we included:
"Possibly, the reliability of the AMATERASS dataset needs to be tested under various climatological conditions. Therefore, ground-based measurements should include stations at high altitude, in the mainland, near the sea, near to urban sites etc. The SKYNET Chiba station can be considered representative of the urban conditions of the mainland region, while the other SKYNET stations can be roughly grouped as representative of a subtropical region possibly affected by desert and continental aerosols. Because of the necessity of examining other climates and different conditions and to extend our validation to the whole Japan, we accomplished further comparisons with respect to 47 stations of the Japanese Meteorological Agency (JMA) surface network of pyranometers, some of them also belong to the Baseline Surface Radiation Network (BSRN), relying on the rigorous quality control performed by JMA. This allowed to distinguish three main climatological regions and additional smaller areas presenting a distinct response."

Then, we further faced this issue in Section 3.2 when discussing the new right panel of the updated Fig. 3 as follows (p. 9, l. 14 to p.10, l. 11):
"The right panel of Fig. 3 shows the correlation between monthly RMSE and cloud fraction at each JMA station for January to December 2016 plotted over the Japan Digital Elevation Model derived

from GTOPO-30 (https://lta.cr.usgs.gov/GTOPO30). Overall, the correlation was always positive but it ranged from more than 0.9 to less than 0.2 for the different stations. As Japan extends from north to south for about 3000 km, it is characterized by a variety of climatic regions which affected the pattern of the correlation. Indeed, according to previous studies (e.g. Ohtake et al., 2015), we can distinguish at least three main climatological regions and additional smaller areas. The first region (enclosed by the blue dashed line) is the norther part of Japan, mostly characterized by a subarctic climate, which provided a uniform response and small differences between Hokkaido and the north of the mainland (i.e. Tohoku). Here, except for two stations located in the Pacific sector of Japan, usually the correlation was lower than 0.4. Then, the large central region (within the red dashed line), which is characterized by humid and temperate climate, presents a more articulated pattern. The flat and strongly urbanized Kanto region (i.e. around Tokyo) showed the highest correlation values (r > 0.9). The SKYNET station of Chiba University is located here and it is supposed to be representative of this area. Then, correlations became slightly lower toward south with an evident distinction between the east coast, characterized by higher correlations, and the west coast, which presents lower values. It is worth noting that in winter the west coast is usually affected by elevate snowfall levels, while the Pacific coast usually shows frequent clear sky conditions. Fukue SKYNET station is located in the south-west sector of this region.

Stations in mountain regions usually present peculiar features that distinguish them from the stations located near the sea. A recent validation of three satellite-based radiation products over an extensive network of 313 pyranometers across Europe (Urraca et al., 2017) showed that stations sited in the Alps and Pyrenees have errors (i.e. RMSE and MB) two or three times larger than the ones of other locations. In mountainous regions, the altitude varies sharply and affects not only surface related parameters, but also the state of the atmosphere. Therefore, satellite models can fail in such areas because the spatial and temporal resolutions are not high enough to account for the sharp terrain and changing weather conditions (Dürr et al., 2010; Castelli et al., 2014). Accordingly, although not shown here, in the central and norther Japan we found larger RMSE values for stations in mountainous regions compared with seaside stations. Finally, we note a somewhat uniform correlation (r > 0.6-0.8) in the subtropical region (enclosed by the green dashed line) along the Pacific Ocean where the largest precipitation usually occurs. The SKYNET stations of Cape Hedo and Miyako are located in this latter region.

The peculiarity of the different regions is manifest when focusing on the annual cycle of RMSE (see inset in the right panel of Fig. 3). Generally, higher RMSE values were found in summer being the values highest in the subtropical region, followed by the central and the norther region. On the other hand, in winter RMSE values in the north were found similar to the ones of the subtropical region. Future analyses, based on longer time series, are expected to further highlight the satellite uncertainties at the different locations and, potentially, improve the accuracy of the satellite-derived dataset by site-adaptation methods (Polo et al., 2016)."

[Figure]

*Fig. 3 [new]: (left panel [updated]) Influence of cloudiness (i.e., total cloud fraction, blue to yellow contour map) on the monthly RMSE of ground observations and Himawari-8 estimates of solar radiation at the 47 stations (points) in the Japanese Meteorological Agency (JMA) network in May 2016; inset: Scatter plot of the total cloud fraction and RMSE for the stations in the central and south regions (i.e. within the area delimited by the black line). (right panel [new]) Correlation between monthly RMSE and cloud fraction at each JMA station for 2016 plotted over the Japan Digital Elevation Model derived from GTOPO-30 (https://lta.cr.usgs.gov/GTOPO30). The three main regions, i.e. north, central and south, are enclosed by blue, red and green dashed lines, respectively; violet circles show the location of the SKYNET stations; inset: mean RMSE (bold lines) and minimum and maximum RMSE (points) for the different regions for December to January 2016.*

The overall evaluation was performed with Skynet, BSRN and other stations from the JMA, so the Title of this paper could be optionally renamed as "... by ground-based measurements".

R -> We changed the title as suggested.

On page 5, line 25 and page 9, line 30, is there a classification of sky conditions in the bibliography based on the CSI? This will be helpful for the readers in order to have a sense of quantification for the various sky conditions.

R -> although to our knowledge there is not such specific classification, using the clear sky index or the clearness index for defining the three main categories of sky conditions (i.e. clear, broken and overcast) is widely accepted. For example, some authors have performed analyses based on the following values: [0.00-0.34] overcast, [0.34, 0.65] broken, [0.65, 1.00] clear (e.g. Serrano et al., 2006; Bech et al., 2015). We included such considerations in the revised manuscript (p. 7, l.16-18).

Page 6, line 4: Mention additionally the minimum solar energy potential in such large SZAs for strengthening this consideration.

R -> We mentioned that for Chiba station the GHI is around 300 and 150 W/m$^2$ at SZA of 70 and 80, respectively.

On page 7, lines 4-5, there is a need for discussion of these results (MB and RMSE) with the mentioned (line 3) or additional references, as to provide direct comparison with similar approaches and satellites.
R -> Following your advice, we extensively compared those as well as other results of the present validation exercise with previous references at both 15 min and 1 h resolution. Note that, in the revised manuscript, we faced this issue in the updated discussion section (p. 17, l. 1-27).

Page 7, line 17: This is an important aspect and it needs a short description of the magnitude of this effect (with reference).
R -> A recent validation of three satellite-based radiation products over an extensive network of 313 pyranometers across Europe (Urraca et al., 2017) showed that stations located in the Alps and the Pyrenees have errors (i.e. RMSE and MB) two or three times larger than the ones obtained in most of the other locations. In such locations, the altitude varies sharply and affects not only surface related parameters, but also the state of the atmosphere. Satellite models fail on mountainous regions because the spatial and temporal resolutions are not high enough to account for the sharp terrain and changing weather conditions (Dürr et al., 2010; Castelli et al., 2014).
Note that we included this paragraph earlier in the new manuscript (p. 9, l. 30 to p. 10, l. 2).

Page 7, line 22: Is there a need for an altitude correction for the satellite estimations?
R -> the negative bias occurs only in winter so it can hardly be connected with some specific altitude-related problem. However, only Nagano station is sited at a slightly relevant altitude (about 400 m a.s.l.) while the other stations are well below, so we cannot check altitudinal issues. On the other hand, issues mentioned in the previous comment likely affected the validation. For example, usually such locations also present a higher RMSE than other locations close to ocean. Potentially, for these locations, site-adaptation methods (Polo et al., 2016) could be important in improving the accuracy of the satellite-derived dataset.

Page 10, line 24: Provide comparable results from the bibliography.
R -> We referred the reader to Fig. 4 of Nottrott and Kleissl (2010) where the authors showed a larger bias (annual mean 18 % and up to about 50 % in summer) when comparing the SUNY modeled dataset with weather stations in California.

Page 11, line 25: What about high aerosol loads without the impact of PW? Huttunen et al. (2014) explains only this (effect of water vapor on the determination of aerosol direct radiative effect). A more focused reference is needed.
R -> We included additional more focused references (p.15, l. 18-21). For example, we reported the results of the study of Papadimas et al. (2012) where the direct radiative forcing efficiency was estimated for the whole Mediterranean basin. They obtained a direct radiative forcing efficiency at the surface of about 100 W/m$^2$ per AOD unit at the 96 % of the locations. Moreover, we also discussed the results of additional studies focused on the fact that the instantaneous forcing efficiency also depends on the different aerosol type as well as on the SZA (Cachorro et al., 2008; Di Biagio et al., 2009, 2010). Overall, these studies reported values roughly in the range 100-150 W/m$^2$ (Cachorro et al., 2008) and 100-200 W/m$^2$ (Di Biagio et al., 2009, 2010).

Percentages of energy attenuation are also welcome (lines 22-23) in order to be useful for pv reduced production because of high aerosol loads.

R -> In the revised version of the manuscript, in the original Fig. 9 (now Fig. 10, see below), we included two additional panels showing the attenuation in percentages. There, we tried to remove the SZA effect on AOD by computing the results in function of the aerosol optical depth slant (AODS i.e., AODS = AOD/cos(SZA), see Garcia et al., 2006) and we obtained -14.3 and -14.5 % per AODS unit at Chiba and Fukue, respectively.

[Figure]

*Fig. 9 [**updated, now Fig. 10**] – (top panels) Scatter plots of the difference in clear-sky surface global radiation (SKYNET observations minus Himawari-8 estimates) and measured SKYNET AOD at 500 nm under clear-sky conditions between January and December 2016 at the Chiba (left panel) and Fukue (right panel) stations. The regression line, its slope, and the correlation coefficient are also shown. (bottom panels) Scatter plots as above but for percentage difference in global radiation and aerosol optical depth slant (AODS).*

Finally, on page 12, lines 4-5, provide comparable results from the bibliography.

R -> these results are roughly consistent with a previous investigation (Kudo et al., 2010) based on ground-based observations and performed in a close location (Tsukuba, distance about 50 km). In this previous study the authors reported a direct aerosol forcing larger for spring to summer (-22 W/m[2]) and small in winter (-8.3 W/m[2]).

In the end of this paragraph the author may provide a short description of the potential EXAM upgrade with the inclusion of the aerosol impact.

R -> In the revised manuscript (p.16, l.29-32), the following short discussion was included: "A recent study showed that the diurnal variation patterns in Himawari-8 AOD data are consistent with those seen in SKYNET observations (Irie et al., 2017). Thus, Himawari-8 aerosol products would provide a unique spatial and diurnal variation information which, once included in the next version of the EXAM algorithm, is expected to reduce this positive bias under clear sky conditions."

The high spatial and temporal resolution of the Himawari-8 satellite in conjunction with

near real-time algorithms like the EXAM, will improve the precision of the solar farms planning and production control with clear benefits for the local energy transmission and distribution system operators. This paper after the above corrections could be a step forward to the efficient and full integration of the natural energy resource to the electricity grid and will contribute to the development of upgraded energy management systems.

R -> thank you!

**New references**

[revised manuscript text omitted]

Anonymous Referee #2

Review of the manuscript "Evaluation of Himawari-8 surface downwelling solar radiation by SKYNET observations" by Damiani et al., 2018.
Summary: The manuscript describes the validation of the solar irradiance dataset from the Himawari-8 satellite by surface based observations using the SKYNET and JMA networks in Japan for the whole year 2016. The solar irradiance dataproduct from Himawari-8 is obtained from the EXAM algorithm. The comparison is performed between surface point observations and satellite derived solar irradiance data ed for different spatial and temporal resolutions. Several influencing parameters such as aerosol optical depth and surface albedo is investigated. The largest part of the paper discusses the quality of the solar irradiance product with respect to varying cloudiness, which is responsible for the largest variation in solar irradiance. The overall objective is to validate solar irradiance retrievals at high temporal and spatial resolution in order to use these datasets in near realtime for informing photovoltaic installations of the expected solar power because rapid transients can be harmful to PV plants, as stated in the manuscript at page 3, line 3.
Comments: The manuscript is well written and has a good structure. The results and conclusions are discussed and presented quite objectively so that the reader can make his own opinion as to the quality of the comparison. The validation of the dataset over only one year of measurements is rather short and might not allow to draw conclusions as to the long-term performance of the satellite-derived solar irradiance product. Nevertheless the results are encouraging and worth to be published as they provide a very good view of the current state-of-the-art in retrieving surface solar radiation from an assimilation of surface based and space-based data on an unprecedented time and spatial resolution.
R -> We thank Referee #2 for his/her valuable comments and suggestions.

In contrast to the authors I conclude from this study that the solar radiation product as presented here is far from reaching the objectives stated in the introduction and which seem to be the rationale for the satellite itself. In general I have the impression that the authors have drawn very optimistic conclusions from their dataset. I suggest modifying the abstract and conclusions to provide a more objective assessment of the satellite solar irradiance product, especially in the frame of the overall objective of using the high temporal solar data for PV applications.

- The all-sky solar radiation data shows large deviations with respect to the surface based datasets, especially at the high temporal resolution of 2.5 minutes. This is very well seen in Figure 6, where deviations between the SKYNET pyranometer and Himawari-8 are strongly dependent on cloudiness (Clear Sky index is used here) with deviations which can exceed a factor of 3! With regard to the stated objective to use these high temporal resolution data for near realtime forecast for PV applications, especially under fast varying gradients induced by changing clouds, I would like to have this aspect more clearly addressed in the discussion and conclusion section of the manuscript.
R -> Because of this comment as well as your comment below (cf. Conclusion, page 13, line 14), first we expanded the introduction of the revised manuscript (p. 3, l. 5-16) by including the results of previous validation exercises obtained from datasets with a resolution at 1 h and shorter (at 15 min). Then, in the updated discussion section (p. 17, l. 1-27), we extensively discussed our findings in the light of these previous results. Overall, we confirmed the good accuracy of AMETERASS at 1 h and at shorter time resolution if compared to previous validations of other state-of-the-art products. Please see our detailed reply following your comment below (cf. Conclusion, page 13, line 14). On the other hand, we agree with the reviewer in that the validation at 2.5 min showed evident deviations and large RMSE. Regrettably, since there is a lack of concurrent operational products of

surface solar radiation with a resolution comparable to AMATERASS, we cannot put our results at 2.5 min within the context of other state-of-the-art products. Nevertheless, it is worth noting that even recent validations at 1 h (e.g. Federico et al., 2017) showed RMSE values comparable with our values at 2.5 min.

Following your suggestion, in both abstract and conclusion of the revised manuscript we highlighted the difficulty of EXAM algorithm in reproducing the observations at 2.5 min.
In the updated abstract, we included the following sentence:
"…However, results depended on the time step used in the validation exercise, on the spatial domain and on the different climatological regions. In particular, the validation performed at 2.5 min showed the largest deviations and RMSE values ranging from about 110 $W/m^2$ for the mainland to a maximum of about 150 $W/m^2$ in the subtropical region."

Then, in the updated conclusions, we stated that:
"Overall, our analysis confirmed the good accuracy of the AMATERASS solar global radiation product at temporal resolution of 1 h and 10 min but showed larger deviations at 2.5 min. An improved algorithm better accounting for the cloud effects would certainly alleviate such deviations which are more evident under fast varying gradients induced by changing clouds. For example, AMATERASS "sees" slightly more clear sky scenes compared with surface observations while it tends to overestimate the solar radiation under cloudiness conditions. A finer satellite spatial resolution would improve the identification of small cumulus clouds.
Moreover, both REs and aerosols contribute to these deviations but their impact is much smaller. Nevertheless, the portion of these deviations, which arises from the intrinsic difference in comparing "punctual" measurements and satellite pixels, would be hardly removed. When the pixel is partly cloudy, surface measurements can greatly vary depending on whether clouds are located or not along the solar beam path to the sensor possibly resulting in a satellite overestimation or underestimation of the irradiance at ground. Due to the large field of view of the pyranometer and depending on the solar zenith angle conditions, even clouds located at many kilometers from the station can potentially affect observations. Unless such effects are accounted by a sufficiently long temporal integration of the surface observations, this would result in discrepancies with respect to satellite-based estimates."

Specific comments:
-Page 5, line 25: The CSI is also a function of the cloud optical depth, not only cloud fraction. A fully overcast sky with cirrus clouds might have a CSI close to 1. Does cloud type have a significant influence on the data product?
R -> As pointed out by the referee, CSI is a function of cloud fraction and cloud optical depth. Nevertheless, using the clear sky index or the clearness index for defining the three main categories of sky conditions (i.e. clear, broken and overcast) is widely accepted (e.g. Serrano et al., 2006; Bech et al., 2015) and proved to be a simple but helpful way to easily compare large datasets of global irradiance under different sky conditions. We included such considerations in the revised manuscript (p. 7, l.16-18). For your reference, we reported below an example of comparison between cloud fraction and CSI for Chiba station. The cloud fraction explains more than 90 % of the variance of CSI index.
Preliminary analysis based on H8 cloud type classification did not show significant influence of specific cloud type on the solar irradiance differences between satellite and ground observations.

[Figure]

*Monthly means in total cloud fraction visual observations (left axis) and clear sky index (right axis, values in reverse order) in 2016 at Chiba station (cloud fraction averaged from 2 near JMA stations).*

-page 6, line 10. I assume this should be irradiance, instead of radiance?
R -> Yes, thank you!

- page 7, lines 13-16 and Figure 3, left inset: I disagree with the statement made by the authors that the data supports the assumption that a larger RMSE is correlated with cloud fraction. The left inset to Figure 3 is made up of two distinct dataclouds, 6 at high cloud fraction, of which 4 are outliers, and only the two leftmost which confirm the authors's assumption. If these 6 datapoints at high cloud fraction are neglected, then the remaining datapoints show no correlation at all between RMSE and cloud fraction. The linear fit is too suggestive and not representative for the dataset.
R -> we agreed with the referee in that the inset does not clearly confirm the correlation between RMSE and cloud fraction so we further extended this analysis, updated the old Fig. 3 and added a new figure in the right panel. Overall, we found that this assumption is not always valid since, in addition to cloudiness, specific features of the different climatic regions as well as topographic constrains likely determined the RMSE. In the revised version of the manuscript the following discussion was included (p.9, l. 14 to p.10, l. 30).

[revised manuscript text omitted]

- page 7, discussion on surface albedo. I have understood that the surface albedo is retrieved from the satellite data itself, and that problems occur when clouds and albedo are misclassified.
My question: Is the surface albedo retrieved independently for every time slot, e.g. every 2.5 minutes, or does the algorithm use the fact that while clouds can very extremely fast, surface albedo will be slowly varying, on the scale of days or more?
R -> yes, the retrieval of the surface albedo is based on a statistical approach over longer periods (30 days time window) which estimates the land surface albedo by the 2-nd minimum reflectance method (Fukuda et al., 2013). We included this information in the revised manuscript (p. 11, l. 1-3).

-page 8, line 17-18. It would the interesting to know more about the dependency between RE and SZA and add some quantitative values.
R -> Following the suggestion of the reviewer, in the revised manuscript we expanded this section (p.12, l. 3-11) by mentioning that previous studies (Piedehierro et al., 2014) found that at middle/low SZA (roughly < 40°) the occurrence of REs is almost equally distributed while for larger SZA (roughly > 40°) their frequency decreases along with increasing SZA.
On the other hand, simulations showed that for the overhead zenith angle the magnitude of the REs is highest for low cloud optical depth (COD, about 2-3), while their magnitude increases with increasing SZA and reach the maximum for high COD values (Pecenak et al., 2016).
In agreement with these previous results (Piedehierro et al., 2014; Pecenak et al., 2016), our dataset shows that, although REs are nearly equally distributed during the day, the occurrence of the strongest events occur mostly at high SZA. We also checked the relation between COD and REs by exploiting H8 COD data (Nakajima & Nakajima, 1995) for Chiba station. In the greatest majority of the cases REs resulted to be coupled with COD of about 0.5-5, although a few were associated with higher COD.

-page 11 and Figure 9. In my opinion the solar radiation forcing with respect to AOD should depend to some extent on airmass, e.g. SZA, since the attenuation of solar radiation is mainly from the direct beam radiation and thus follows the Beer-Lambert law.
R -> Yes, the instantaneous forcing efficiency depends also on airmass i.e. SZA.
In the revised version of the manuscript (p.15, l.29 to p.16, l.11) we clarified this issue. We added the discussion below, we provided new references and updated the original Fig. 9 with two additional panels showing the attenuation in percentages per aerosol optical depth slant (AODS) unit.

"In a recent paper (Irie et al., 2017), by running radiative transfer simulations including aerosol proprieties observed at the SKYNET Chiba station, we showed that the relative decrease of solar radiation induced by an increasing AOD is larger at high SZAs. In some previous studies based on surface measurements (e.g. Xia et al., 2007; Cachorro et al., 2008; Di Biagio et al., 2009, 2010), the instantaneous forcing efficiency has been computed for small SZA ranges as a preliminary step to estimate the mean direct aerosol forcing. However, usually small differences in the forcing efficiency have been reported for SZA of 10-60°. Thus, because of the additional constrains due to the satellite acquisition times and clear sky observations and since data recorded at high SZAs were previously excluded, in Fig. 10 we showed the forcing efficiency for the whole dataset. An additional issue is due to the fact that the instantaneous forcing efficiency can be only partially described by a simple

linear fit and, for the same aerosol family, the estimation of the change in solar radiation for AOD unit is rather different if computed over different AOD ranges. In our dataset, while we have a large range of AOD for mid SZAs, we have a much smaller range for high SZAs which is probably insufficient to extrapolate a reliable efficiency. Moreover, the slope of the linear regression is also affected by the different type of aerosols (e.g. Di Biagio et al., 2009, 2010), therefore sampling the data for different SZA could introduce further issues if episodes of different aerosol types occurred along the year. Nevertheless, in the bottom panels of Fig. 10 we tried to remove the SZA effect on AOD by computing the percentage decrease of solar radiation in function of the aerosol optical depth slant (AODS i.e. AODS = AOD/cos(SZA); e.g. Garcia et al., 2006) and we obtained -14.3 and -14.5 % per AODS unit at Chiba and Fukue, respectively."

[Figure]

*Fig. 9 [**updated, now Fig. 10**] – (top panels) Scatter plots of the difference in clear-sky surface global radiation (SKYNET observations minus Himawari-8 estimates) and measured SKYNET AOD at 500 nm under clear-sky conditions between January and December 2016 at the Chiba (left panel) and Fukue (right panel) stations. The regression line, its slope, and the correlation coefficient are also shown. (bottom panels) Scatter plots as above but for percentage difference in global radiation and aerosol optical depth slant (AODS).*

- Conclusion, page 13, line 14. I do not see how this statement is an outcome of this paper, since the only comparison to an other satellite product was Himawari-7.
R -> we removed this sentence.
However, in the light of your comment, we explicitly compared our findings with results from recent validation exercises performed for other satellites at high temporal resolutions. Therefore, additional validation references have been reported in the introduction section of the revised manuscript (e.g. Ruf et al., 2016; Zo et al., 2016; Qu et al., 2017; Kosmopoulos et al., 2018). Then, an explicit comparison with these and other previous results has been included in the new discussion section (p. 17, l. 1-27) as follows:

"Since the different weather conditions finally determine the validation, it is not straightforward to compare our results with validations obtained by previous studies based on other satellites/algorithms and regions. Nevertheless, our results appear to be at least comparable to the previous ones. Since there is a lack of concurrent operational products of surface solar radiation with a resolution comparable to AMATERASS (i.e. at 2.5 min), in the introduction we focused on results obtained from datasets with a resolution of 1 h and 15 min. Here we note that the majority of the earlier studies (e.g. Nottrott and Kleissi, 2010; Thomas et al., 2016a,b; Qu et al., 2017) reported a general overestimation in the estimated solar radiation by a few tens of W/m2. This occurred despite the fact that aerosols were taken into account in the algorithm. A comparable overestimation, which did not depend on the temporal resolution adopted, has been found in our analysis. This suggests that, when dealing with all-sky conditions, usually the role of aerosols is marginal compared with the one of cloudiness. In particular, in correspondence of stations located in the flat and strongly urbanized Kanto region (e.g. Chiba), the satellite overestimation ranged from a few percent in winter to about 10 % in summer (with a MB around 20 W/m2 on annual basis) and roughly followed the annual cycle of the cloud amount (i.e. smaller in winter and larger in summer). On the other hand, the RMSE between observations and satellite-based estimates did depend on the resolution (see Tab. 2) and in our analysis the annual RMSE resulted to be reduced to about one third of its original value (computed from data at 2.5 min) when statistics were computed from daily means.

For Japan, RMSE values computed from high resolution (i.e. 10 minutes) estimates usually ranged between 60 and 90 W/m2 with larger values occurring in summer, especially in the subtropical region (see the inset in the right panel of Fig. 3). Such values resulted to be comparable to RMSE values obtained from validations performed at similar resolution for Europe (i.e. 15 min, Qu et al., 2017) or even to values from hourly datasets for Canada and Italy (Djebbar et al., 2012; Federico et al., 2017). In the latter evaluation, the authors reported a RMSE in the range of 150-200 W/m2 for three mountain stations (two of them over 2000 m a.s.l.). It is worth noting that such values are higher even than RMSE obtained from the instantaneous (i.e. 2.5 min) dataset at tropical stations. In this recent study (i.e. Federico et al., 2017) the authors suggested that in these locations RMSEs were affected by specific constrains induced by the morphology and/or the larger amount of cloudy days with respect to other locations. In Japan, among the SKYNET and JMA stations not one is located at analogous altitudes so we cannot compare this feature. On the other hand, we checked if the largest RMSE values found for the subtropical islands were associated with less clear sky days. By using the CSI between 0.9 and 1 as a proxy of clear sky conditions, it resulted 5 % more clear sky days at Chiba than in the other subtropical SKYNET locations. Therefore, this could explain the slightly worse results obtained for the SKYNET subtropical stations compared to Chiba."

Figures: Figure 2: The resolution is not very good and the values shown in the subsets are not easy to read.
R -> In the revised paper we included high resolution figure files and now the text should be readable; please also note that the statistical values in the subset are also reported in Tab. 1.

Figure 3: As mentioned in my previous comment, the suggestive linear fit between RMSE and cloud fraction is not convincing, since it depends on the two outlier points at high RMSE and high cloud fraction. Another view could be that there the top six points are outliers, and the remaining points show no clear correlation between RMSE and cloud fraction.
R -> we carefully addressed this comment in the previous reply. Overall, we found that, in addition to cloudiness, also climatic regions and topographic constrains determine the correlation. So, we updated the left panel and included a new analysis in the right one.

Figure 6: The text in the four sub figures is very difficult to read, possibly for lack f resolution, and the chice of colors. Especially the yellow text is unreadable. The variables need to be defined in the caption.

R -> In the revised paper we included high resolution figure files and now the text should be readable. We also defined the variables in the caption.

Figure 9: as mentioned previously, I expect the aerosol radiative forcing to also depend on airmass. Is it possible to add this information, or mention it in the accompanying text?

R -> Agreed! We included this information in the revised manuscript (see our previous reply).

Figure 10: I suggest to plot the right (red) AOD axis in reverse, to show even more clearly the anti-correlation between AOD and residuals.

R -> we tried that but, in this way, we cannot distinguish the error bars of the two datasets, so finally we did not modify the original figure.

In the caption, the error bars need to be better explained. Does it for example represent the standard deviation?

R -> Yes, it is the standard deviation. In the revised manuscript, we explicitly mentioned this in the caption.

Table 2: I assume that the information given in table 2 should be consistent with the information given in Figure 6? In that case, it would be better to harmonise this information, and give the slope in the same units, not the inverse.

R -> actually the information given in table 2 is consistent with the information given in Figure 6. Nevertheless, while in Fig. 6 we reported the statistics for different clear-sky index ranges, in table 2 we show the overall statistics related to the whole dataset (i.e., independently on the cloud conditions).

**New references**

[revised manuscript text omitted]